# ON THE INFLUENCE OF SHAPE, TEXTURE AND COLOR FOR LEARNING SEMANTIC SEGMENTATION

## ABSTRACT

In recent years, a body of works has emerged, studying shape and texture biases of off-the-shelf pre-trained deep neural networks (DNN) for image classification. These works study how much a trained DNN relies on image cues, predominantly shape and texture. In this work, we switch the perspective, posing the following questions: What can a DNN learn from each of the image cues, i.e., shape, texture and color, respectively? How much does each cue influence the learning success? And what are the synergy effects between different cues? Studying these questions sheds light upon cue influences on learning and thus the learning capabilities of DNNs. We demonstrate that the way DNNs perceive the world can be broken down into distinct sources of evidence. We study these questions on semantic segmentation which allows us to address our questions on pixel level. To conduct this study, we develop a generic procedure to decompose a given dataset into multiple ones, each of them only containing either a single cue or a chosen mixture. This framework is then applied to two real-world datasets, Cityscapes and PASCAL Context, and a synthetic data set based on the CARLA simulator. We learn the given semantic segmentation task from these cue datasets, creating cue experts. Early fusion of cues is performed by constructing appropriate datasets. This is complemented by a late fusion of experts which allows us to study cue influence location-dependent on pixel level. Our study on three datasets reveals that neither texture nor shape clearly dominate the learning success, however a combination of shape and color but without texture achieves surprisingly strong results. Our findings hold for convolutional and transformer backbones. In particular, qualitatively there is almost no difference in how both of the architecture types extract information from the different cues.

## 1 INTRODUCTION

Visual perception relies on visual stimuli, so-called visual cues, providing information about the perceived scene. Visual environments offer multiple cues about a scene, and observers need to assess the reliability of each cue to integrate the information effectively Jacobs (2002). To recognize and distinguish objects, for example, their shape and texture provide complementary cues. In cognitive science and psychology the influence of different cues and their combinations for human perception is addressed in various studies Bankieris et al. (2017); Jacobs (2002); Michel & Jacobs (2008). Given the widespread use of deep neural networks (DNNs) for automatically extracting semantic information from scenes, it is important to investigate 1) how these models process and rely on different types of cue information, as well as 2) what they are able to learn when only having access to specific cues. With respect to the first question several hypotheses about dominating cue exploitation (cue biases) of trained convolutional neural networks (CNNs) were formulated and supported by experimental results for image classification tasks Geirhos et al. (2018); Islam et al. (2021); Tuli et al. (2021). Early in the evolution of CNNs a shape bias was hypothesized, stating that representations of CNN outputs seem to relate to human perceptual shape judgement Kubilius et al. (2016). Even though this suggests that CNNs tend to base their prediction on shape information, this is only valid on a local perspective Baker et al. (2018) and not an intrinsic property Hosseini et al. (2018). On the contrary, multiple studies were performed on ImageNet-trained CNNs Deng et al. (2009), indicating that those CNNs have a bias towards texture Geirhos et al. (2018); Baker et al. (2018); Brendel & Bethge (2018). To reveal biases in trained DNNs, cue conflicts are often generated Geirhos et al.

| original image | shape | texture | pixel color |

Figure 1: A sample of cues and cue combinations extracted from the Cityscapes dataset, based on which cue expert models are trained.

(2018); Gavrikov et al. (2024). To this end, a style transfer with a texture image of one class, e.g. showing an animal's fur or skin, is applied to an ImageNet image representing a different class.

In summary, previous studies 1) mostly study the biases of trained DNNs w.r.t. to image cues and their influence on the networks robustness Geirhos et al. (2018); Kamann & Rother (2020); Naseer et al. (2021); Qiu et al. (2024), 2) often rely on style transfer or similar image manipulations to test for biases Li et al. (2020); Islam et al. (2021); Dai et al. (2022), and 3) largely focus on image classification DNNs Brendel & Bethge (2018); Hermann et al. (2020); Tuli et al. (2021).

In this work, we present a study that is novel in all three aspects. 1) We switch the perspective, studying how much influence different image cues (and arbitrary combinations of those) have on the learning success of DNNs. 2) We do not rely on style transfer but rather utilize and develop a set of methods, combining them into a generic procedure to derive any desired cue combination of shape, texture and color from a given dataset, cf. fig. 1. 3) We lift our study to the task of semantic segmentation. This opens up paths to completely new studies as it allows for the decomposition of datasets into arbitrary combinations of cues and enables more fine-grained analyses such as image-location-dependent cue influences. Our study setup serves as basis to train expert networks exclusively relying on a specific cue or cue combination. We perform an in-depth behavioral analysis of CNNs and transformers on three different semantic segmentation datasets, namely Cityscapes Cordts et al. (2016), PASCAL Context Mottaghi et al. (2014) and a synthetic one recorded with the CARLA driving simulator Dosovitskiy et al. (2017). Our study brings the different cue and cue combination influences on DNN learning into a consistent and intuitive but prior to this not proven order. It turns out that neither texture nor shape clearly dominate in terms of learning success. However, a combination of shape and color achieves surprisingly strong results. These findings hold for CNNs and transformers. In particular, qualitatively there is almost no difference in how both architecture types extract information from the different cues, i.e., the choice of backbone has almost no impact on the order of the cue influences. Additionally, by a pixel-wise late fusion of cue (combination) experts, we study the role of each pixel for cue influences, showing quantitatively that small objects and pixels at object borders are dominantly better predicted by shape experts. Our contributions are summarized as follows:

- We provide a generic procedure to derive cue combination datasets from a given semantic segmentation dataset. In particular, we provide a method to derive texture-only datasets.

- Our general setup allows to study disentangled image cues, down to the detail degree of brightness-only. We perform an in-depth analysis with up to 14 learned cue combination experts per dataset and several late fusion models, contributing the first cue influence study in semantic segmentation. Additionally, we include transformers that are so far underresearched in the broader context of cue influences.

- While previous studies report strong biases of ImageNet-trained DNNs towards texture under style transfer, our study reveals that for real-world data shape and texture are equally important cues for successful learning. The role of the cues varies across classes and image location, but not w.r.t. choosing between CNNs and transformers.

Our code including the data generation procedure is publicly available at TBA.

## 2 RELATED WORK

Cue biases and cue decompositions are pivotal concepts in understanding how neural networks interpret visual information. We start by reviewing existing approaches of cues bias analyses in classification, followed by methods to decompose or manipulate data to allow for investigations on different cues. We conclude by addressing the underexplored area of texture and shape biases in semantic segmentation.

**Shape and Texture Biases in Image Classification.** DNNs learn unintended cues that help to solve the given task but limit their generalization capability Geirhos et al. (2020). In Geirhos et al. (2018) it was measured whether the shape or the texture cue dominates the decision process of an ImageNet-trained classification CNN by inferring images with conflicting cues (e.g. shape of a cat combined with the skin of an elephant), revealing that ImageNet pre-training leads to a texture bias. Technically, Geirhos et al. (2018) stylize images via style transfer to create cue conflict images. This was also adopted by Li et al. (2020) and Islam et al. (2021). In the latter work, the bias is computed on a per-neuron level. Hermann et al. (2020) showed that the selection of data and the nature of the task influence the cue biases learned by a CNN. Experiments in Naseer et al. (2021); Tripathi et al. (2023); Tuli et al. (2021); Geirhos et al. (2021) demonstrate that transformers exhibit a shape bias in classification tasks, which is attributed to their content-dependent receptive field Naseer et al. (2021). Our focus differs in two key aspects: 1) we study cue influences on learning success rather than biases, 2) we consider the task of semantic segmentation instead of image classification.

**Data Manipulation for Bias Investigation.** To examine the influence of cues, datasets are manipulated by either artificially combining cues to induce conflicts Baker et al. (2018); Geirhos et al. (2018); Tripathi et al. (2023); Theodoridis et al. (2022) or by selectively removing specific cues from the data Dai et al. (2022); Zhang & Mazurowski (2024). To remove all but the shape cue, edge maps Baker et al. (2018); Mummadi et al. (2020); Tripathi et al. (2023), contour maps Baker et al. (2018), silhouettes Baker et al. (2018) or texture reduction methods Dai et al. (2022); Heinert et al. (2024) are used. Patch shuffling has been proposed to remove the shape cue but preserve the texture Brendel & Bethge (2018); Luo et al. (2020); Dai et al. (2022). In contrast to classification, data preparation is more complex for semantic segmentation as multiple objects and classes are present in the input which differ in their cues. In particular semantic segmentation on texture-only data is challenging Cote et al. (2023). Removing shape by dividing and shuffling an image as used by Dai et al. (2022) for classification compromises semantic integrity of the image and disrupts the segmentation task. Therefore, we propose an alternative method for extracting texture from the dataset, which is sufficiently flexible to generate new segmentation tasks using in-domain textures from the original dataset.

**Shape and Texture Biases in Image Segmentation.** Up to now, a limited number of works studied shape and texture biases beyond image classification. In Li et al. (2020), for image classification, images are stylized using a second image from the same dataset. For semantic segmentation, only a specific object rather than the full image is used as texture source to perform style transfer. The stylized data is added to the training data to debias the CNN and increase its robustness. Similarly, Theodoridis et al. (2022) stylize data to analyze the robustness of instance segmentation networks under randomized texture. Additionally, an object-centric and a background-centric stylization are used for this study. In Zhang & Mazurowski (2024), the change in shape bias of semantic segmentation networks is studied under varying cues in data. The experiments on datasets with a limited number of images and classes reveal that CNNs prioritize non-shape cues if multiple cues are present. Kamann & Rother (2020) propose to colorize images with respect to the class IDs to reduce the influence of texture during training to prevent texture bias and thereby improving robustness of the trained model. In Heinert et al. (2024), an anisotropic diffusion image processing method is used for removing texture from images. Based on that the texture bias is studied and also reduced. All works mentioned in this paragraph study or reduce biases of DNNs in image segmentation. To the best of our knowledge, the present work is the first one to switch the perspective and study the influence of cues and cue combinations on learning success in semantic segmentation DNNs. This is done on complex datasets with at least 15 classes, by which we obtain different insights.

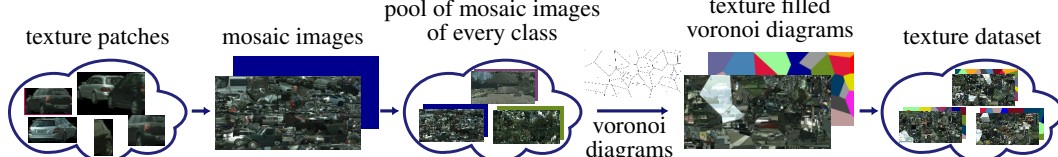

Figure 2: Extraction process of the texture (T) cue. It consists of the three main steps: class-wise patch extraction, class-wise mosaic image construction and segmentation dataset creation based on Voronoi diagrams.

## 3  CUE DECOMPOSITION AND CUE EXPERT FUSION

In this section we introduce the different methods that extract image cues and cue combinations from a given base dataset (original dataset with all cues), from which we train cue and cue combination experts. Technical and implementation details of the methods are provided in appendix A.2.

**Color (C) Cue Extraction.**  Most cue extractions presented in this work modify the base dataset. In order to isolate the color information, we do not modify the base dataset but constrain the cue expert, i.e., the neural network, to process a single pixel's color values via $(1 \times 1)$-convolutions. This prevents the model from learning spatial patterns such as shapes or textures. Furthermore, we decompose color into two components: its gray value (V) and its chromatic value (HS) by switching to the HSV color space and discarding the value channel. Gray values are obtained by averaging or maximizing RGB channels to extract the degree of darkness or lightness of a given color. In what follows, we use the shorthand **C=V+HS** to refer to the respective cues.

**Texture (T) Cue Extraction.**  The texture dataset is constructed in three main steps. 1) For all images in the base dataset texture is extracted by isolating individual segments for each respective class using the semantic segmentation masks to identify and define the boundaries of these segments for cropping. This generates a pool of texture patches per class. 2) For each class a pool of 'mosaic' image is created by randomly stitching texture patches extracted in 1) for one class in an overlapping manner until no texture-free space is left. 3) A new semantic segmentation task, serving as surrogate task, is constructed via Voronoi diagrams. Each cell is uniformly at random assigned a class of the base dataset and filled with a cutout of a random mosaic image corresponding to that class. This step is repeated until a chosen number of Voronoi diagrams are generated, e.g., as many as in the base dataset. Figure 2 illustrates this process.

**Shape (S) Cue Extraction.**  The shape of an object can be described in multiple ways Feldman (2024), defined by the object with unrecognizable/removed texture or by the object defining edges. To this end, we consider two shape extraction methods: Holistically-nested edge detection (HED) Xie & Tu (2015) and a variation of Edge Enhancing Diffusion (EED) Heinert et al. (2024). HED approximately extracts the object defining edges based on a fully convolutional network which has learned to predict a dense edge map through end-to-end training and nested multiscale feature learning. HED approximates the **S** cue. In contrast, EED diminishes texture through diffusion along small color gradients in a given image. The diffusion process follows a partial-differential-equation formulation. As it preserves color, it extracts the cue combination **S+V+HS**. By gray-scaling the diffused image, the cues **S+V** are obtained and and analogously to the treatment described in Color Cue Extraction by switching to the HSV color space and discarding the value channel extracts the cues **S+HS**.

**Texture-free Image Data from CARLA ($S_{rmv}$).**  In the case of simulated data where access to the rendering engine is granted, a nearly texture-free environment can be generated. The open-source simulator CARLA Dosovitskiy et al. (2017) was employed to generate a virtual environment devoid of texture. We replaced all objects' textures by a gray checkerboard, a default texture pattern in CARLA. In addition, weather conditions were set to 'clear noon' to obtain a uniform appearance of the sky. Strongly subsampled video sequences were recorded from an ego perspective in gray scale. This procedure provides an additional way to obtain the cues **S+V** in CARLA.

Table 1: Overview of cues and cue extraction methods. The included cues are S = shape, T = texture, V = gray component of the color and HS = hue and saturation component of the color. Orig is replaced by a shorthand of the respective base dataset.

| shorthand | included cues | description |
|---|---|---|
| all cues | S + T + V + HS | original images / all cues |
| $\text{Orig}_{HS}$ | S + T    + HS | all but gray cues |
| $\text{Orig}_{V}$ | S + T + V | all but chromatic cues |
| $T_{RGB}$ | T + V + HS | texture with color; shape removed |
| $T_{HS}$ | T    + HS | texture with hue and saturation only |
| $T_{V}$ | T + V | texture with grayness only; |
| $S_{EED-RGB}$ | S    + V + HS | shape with color via smoothing texture by edge enhancing diffusion |
| $S_{EED-HS}$ | S    + HS | shape with hue and saturation by edge enhancing diffusion |
| $S_{EED-V}$ | S    + V | shape with grayness by grayscaled edge enhancing diffusion results |
| $S_{HED}$ | S | shape only via contour map by holistically-nested edge detection |
| $S_{rmv}$ | S    + V | shape with grayness via texture removal in the CARLA simulation |
| RGB | V + HS | complete color component of an RGB image, pixel-wise |
| HS | HS | hue and saturation of an RGB image, pixel-wise |
| V | V | gray component of an RGB image, pixel-wise |
| no info | | no information about the data is given representing the absence of all cues |

**Remaining Cue Combinations and Summary.** The remaining cue combinations are obtained as follows: **S+T+V+HS**, **S+T+HS** and **S+T+V** are obtained by treating an original image analogously to the color cue extraction, i.e., by transforming it into HSV color space and projecting accordingly. Analogously, we can process texture images to obtain **T+HS** as well as **T+V**. To represent the complete absence of cues, i.e. no information is given, we consider the performance of randomly initialized DNNs. An overview of all cue expert (datasets) including additional shorthands, encoding the method used and the cue extracted, are summarized in table 1. Note that, in what follows we rely heavily on the shorthands. For additional technical details and exemplary images of the cue extraction methods, we refer the reader to appendices A.1 and A.2.

**Late Fusion of Cue Experts.** Fusing the cue experts' softmax activations serves as an assessment of the reliability of the different cues to combine the information of multiple cues effectively.

# 4 EXPERIMENTS

To analyze the cue influence in learning semantic segmentation tasks, we train all cue and cue combination experts on three different base datasets introduced below and ranging from real-world street scenes over synthetic street scenes to diverse in- and outdoor scenes. We perform an in-depth analysis and comparison of several cue (combination) experts across these three datasets, across different evaluation granularities ranging from dataset-level over class-level to pixel-level evaluations. This is complemented by a comparison of CNN and transformer results. Additional experimental studies and qualitative examples are provided in appendix A.3.

## 4.1 BASE DATASETS & NETWORKS

**Cityscapes.** Cityscapes Cordts et al. (2016) is an automotive dataset which consists of high-resolution images of street scenes from 50 (mostly German) cities with pixel annotations for 30 classes, respectively. There are 2,975 fine labeled training and 500 validation images. As common practice we use only the 19 common classes and the validation data for testing. For model selection purposes we created another validation set from a subset of the 20,000 coarsely annotated images of Cityscapes by pseudo-labelling that subset using an off-the-shelf DeepLabV3+ Chen et al. (2018).

**CARLA.** We generated a dataset using the open-source simulator CARLA Dosovitskiy et al. (2017), version 0.9.14, containing multiple enumerated maps of which we used the towns 1–5 and 7 of similar visual detail level for data generation. We record one frame per second from an ego perspective while driving with autopilot through a chosen city, accumulating 5,000 frames per city in total. For comparability to Cityscapes, we reduced the set of considered classes to 15: *road*, *sidewalk*, *building*, *wall*, *pole*, *traffic lights*, *traffic sign*, *vegetation*, *terrain*, *sky*, *person*, *car*, *truck*, *bus*, *guard rail*. The classes 'bicycle', 'rider' and 'motorcycle' are excluded for technical reasons

because these actors were problematic for the autopilot mode in CARLA. To ensure a location-wise disjoint training and test split, we use town $1$ and $5$ for testing only, whereas $2, 3, 4$ and $7$ are used for training. This results in $20,000$ training images and $10,000$ test images. For model selection purposes, we randomly split of 10% of the training images for validation.

**PASCAL Context.** As a third dataset we analyze cue influences on the challenging PASCAL Context dataset Mottaghi et al. (2014). It provides annotations for the whole images of PASCAL VOC 2010 Everingham et al. (2010) in semantic segmentation style. The images are photographs from the flickr[1] photo-sharing website recorded and uploaded by consumers covering diverse indoor and outdoor scenes Everingham et al. (2010). We consider 33 labels that are contextual categories from Mottaghi et al. (2014). In our experiments we use $4,996$ training images, $2,042$ validation images and $3,062$ test images where the training images of Mottaghi et al. (2014) are split into training and validation images and the original validation images are utilized as test images.

**Neural Networks and Training Details.** For better comparison and to ensure that only the intended cues are learned, all neural networks are trained from scratch multiple times with different random seeds. For all cue (combination) expert models, except for the color cue ones, we employ DeepLabV3 with a ResNet18 backbone Chen et al. (2017); He et al. (2016) (15.9 million learnable parameters) and a SegFormer model with B1 backbone Xie et al. (2021) (13.7 million learnable parameters). Both models have similar learning capacity and achieve similar mean IoU (mIoU) Jaccard (1912) segmentation accuracy. For the color experts, the receptive field is constrained to one pixel, achieved through a fully convolutional neural network with two to three $(1 \times 1)$-convolutions (tuned to achieve maximal mIoU). We trained the CNN models with the Adam optimizer for $200$ epochs and a poly-linear learning rate decay with an initial learning rate of $5 \cdot 10^{-4}$. The transformers were trained with the default optimizer and inference settings as specified in the MMSegmentation library MMSegmentation Contributors (2020) except that we increased the number of gradient update steps to $170,000$ iterations. Note that, to not mix cues unexpectedly, we restricted data augmentation to cropping and horizontal flipping and refrained from augmentations like color jittering. For the late-fusion approach, we use an even smaller version of DeepLabV3 by limiting the backbone to two instead of four blocks to prevent overfitting to the simpler task. The weights of the late fusion models are initialized randomly within a uniform distribution of $[-10^{-3}, 10^{-3}]$.

**Evaluation Protocol.** Cue influence is measured by evaluating the different cue (combination) experts in terms of (m)IoU on the test images across the three different base datasets. To ensure input compatibility, we evaluate experts with a reduced number of channels like the $T_V$ expert on the corresponding $\text{Orig}_V$ data. Comparing the performance drop (gap) between the reduced model and the model trained on the original RGB images (all cues) demonstrates the significance of the removed cues in terms of their impact on the original semantic segmentation task. We use mIoU to capture the performance of all classes equally as, in particular the rare classes often represent vulnerable road users in automotive driving datasets. For the sake of completeness, we report frequency weighted IoU values in the appendix.

### 4.2 NUMERICAL RESULTS

**Cue Influence on Mean Segmentation Performance.** In this section, we analyze the general cue influence in terms of mean segmentation performance. The corresponding results are presented in tables 2 and 3. Firstly, we note a certain expected consistency in the presented mIoU results across base datasets. C experts are mostly dominated by T experts as well as S experts, and those are in turn dominated by S+T experts. The S+T+V expert is the only one getting close to the 'expert' trained on all cues. Nevertheless, the influence of the specific cue (combinations) for the learning success in terms of rankings varies slightly across datasets, however not drastically. These changes are particularly pronounced when comparing domains with greater difference in their characteristics, such as synthetic vs. real-world data. Furthermore, we observe across all three datasets that the RGB version of EED providing the cues S+C achieves surprisingly high mIoU values compared to its S+V counter part, when evaluated on the corresponding original image. This indicates that color and shape in absence of texture encode enough information for a DNN to predict a decent segmentation mask for the original data. However, this mIoU value is dominated by S+T+V as well

---

[1] https://www.flickr.com

Table 2: Cue influence in terms of mIoU performance drop on Cityscapes for DeepLabV3 with ResNet backbone and SegFormer. Cue description follows the listing in table 1. MIoU gaps to maximal performance are stated in percent points (pp.). The abbreviation "City" refers to Cityscapes.

| | S | T | Color V | Color HS | CNN mIoU (%,↑) | CNN gap (pp.,↓) | transformer mIoU (%,↑) | transformer gap (pp.,↓) | change in rank w.r.t. CNN |
|---|---|---|---|---|---|---|---|---|---|
| no info | | | | | $0.25 \pm 0.35$ | 64.97 | $0.33 \pm 0.47$ | 66.02 | → |
| V | | | ✓ | | $6.39 \pm 0.04$ | 58.83 | | | |
| HS | | | | ✓ | $9.33 \pm 0.18$ | 55.89 | | | |
| RGB | | | ✓ | ✓ | $11.31 \pm 0.52$ | 53.91 | | | |
| $S_{HED}$ | ✓ | | | | $13.38 \pm 2.00$ | 51.84 | $11.31 \pm 1.95$ | 55.05 | → |
| $T_V$ | | ✓ | ✓ | | $17.85 \pm 1.30$ | 47.37 | $29.02 \pm 0.31$ | 37.33 | → |
| $S_{EED-HS}$ | ✓ | | | ✓ | $19.48 \pm 3.19$ | 45.74 | $30.93 \pm 0.64$ | 35.42 | ↘1 |
| $T_{RGB}$ | | ✓ | ✓ | ✓ | $20.10 \pm 0.98$ | 45.12 | $31.88 \pm 0.38$ | 34.47 | ↘1 |
| $T_{HS}$ | | ✓ | | ✓ | $20.63 \pm 1.41$ | 44.59 | $29.49 \pm 0.50$ | 36.86 | ↗2 |
| $S_{EED-V}$ | ✓ | | ✓ | | $27.86 \pm 3.17$ | 37.36 | $39.01 \pm 0.75$ | 27.34 | → |
| $S_{EED-RGB}$ | ✓ | | ✓ | ✓ | $42.22 \pm 2.13$ | 23.00 | $50.48 \pm 0.55$ | 15.87 | → |
| $City_{HS}$ | ✓ | ✓ | | ✓ | $59.89 \pm 0.74$ | 5.33 | $58.79 \pm 0.40$ | 7.56 | → |
| $City_V$ | ✓ | ✓ | ✓ | | $64.21 \pm 0.60$ | 1.01 | $64.47 \pm 0.21$ | 1.88 | → |
| all cues | ✓ | ✓ | ✓ | ✓ | $65.22 \pm 0.47$ | 0.00 | $66.35 \pm 0.29$ | 0.00 | → |

Table 3: Cue influence measured in terms of mIoU performance on the synthetic CARLA dataset and PASCAL Context (short: PASCAL). MIoU gaps to maximal performance are stated absolute in percent points (pp.).

| | CARLA mIoU (%,↑) | gap (pp.,↓) | rank change w.r.t. Cityscapes CNN | | PASCAL Context mIoU (%,↑) | gap (pp.,↓) | rank change w.r.t. Cityscapes CNN |
|---|---|---|---|---|---|---|---|
| no info | $0.38 \pm 0.44$ | 75.13 | → | no info | $0.11 \pm 0.11$ | 45.34 | → |
| V | $6.01 \pm 0.08$ | 69.50 | → | V | $2.27 \pm 0.04$ | 43.18 | → |
| $S_{HED}$ | $11.17 \pm 0.65$ | 64.34 | ↗2 | HS | $3.35 \pm 0.10$ | 42.10 | → |
| HS | $14.88 \pm 0.38$ | 60.63 | ↘1 | $S_{HED}$ | $4.71 \pm 1.15$ | 40.74 | ↗1 |
| RGB | $15.77 \pm 0.57$ | 59.74 | ↘1 | RGB | $4.91 \pm 0.05$ | 40.54 | ↘1 |
| $S_{textureless}$ | $26.60 \pm 1.76$ | 48.91 | | | | | |
| $S_{EED-V}$ | $37.25 \pm 1.76$ | 38.26 | ↗4 | $T_{HS}$ | $11.39 \pm 0.36$ | 34.06 | ↗3 |
| $S_{EED-HS}$ | $44.78 \pm 0.85$ | 30.73 | → | $T_{RGB}$ | $17.75 \pm 0.82$ | 27.70 | ↗1 |
| $T_V$ | $46.11 \pm 2.73$ | 29.40 | ↘2 | $S_{EED-HS}$ | $17.80 \pm 1.67$ | 27.65 | ↘1 |
| $T_{HS}$ | $52.66 \pm 1.46$ | 22.85 | → | $T_V$ | $18.43 \pm 0.47$ | 27.02 | ↘3 |
| $T_{RGB}$ | $55.89 \pm 1.90$ | 19.62 | ↘2 | $S_{EED-V}$ | $25.80 \pm 2.40$ | 19.65 | → |
| $S_{EED-RGB}$ | $61.46 \pm 1.03$ | 14.05 | → | $S_{EED-RGB}$ | $31.32 \pm 0.82$ | 14.13 | → |
| $CARLA_{HS}$ | $70.34 \pm 1.56$ | 5.17 | → | $PASCAL_{HS}$ | $36.10 \pm 0.34$ | 9.35 | → |
| $CARLA_V$ | $73.17 \pm 5.19$ | 2.34 | → | $PASCAL_V$ | $45.39 \pm 0.71$ | 0.06 | → |
| all cues | $75.51 \pm 1.50$ | 0.00 | → | all cues | $45.45 \pm 0.18$ | 0.00 | → |

as S+T+HS. Note that a combination of S+T only is impossible since T cannot exist without some kind of brightness or color cue, i.e., V and/or HS.

An additional surprise might be that HED, representing the S cue, reaches only very low mIoU, showing consistently weak performance across all datasets. At the first glance, this is in contrast to human vision since an HED image seems almost enough for a human to estimate the class of each segment in an HED image, cf. fig. 1. It should be noted that each expert is trained on its specific cue and then tested on an original input image from the given base dataset. When alternatively applying a given experts cue extraction technique as pre-processing to real-world images, like Cityscapes or PASCAL Context, HED (with HED pre-processing) surpasses EED (with EED-pre-processing) distinctly and achieves $55.80\% \pm 0.59$ percent points (pp.) mIoU compared to $48.47\% \pm 0.45$ pp. by EED on Cityscapes. On CARLA, EED and HED are on par, reaching an mIoU of $65.93\% \pm 0.72$ pp. and $63.33\% \pm 1.11$ pp. respectively. A comprehensive study of this domain-shift-free cue evaluation is given in the appendix.

**Cue Influence on Different Semantic Classes.** Figure 3 provides an evaluation for Cityscapes, comparing a shape expert based on colored EED images providing the cues S+C and another expert trained on colored Voronoi cells of class-specific stitched patches, providing the queues T+C. This evaluation is broken down into IoU values over the 19 Cityscapes classes. In a visual inspection,

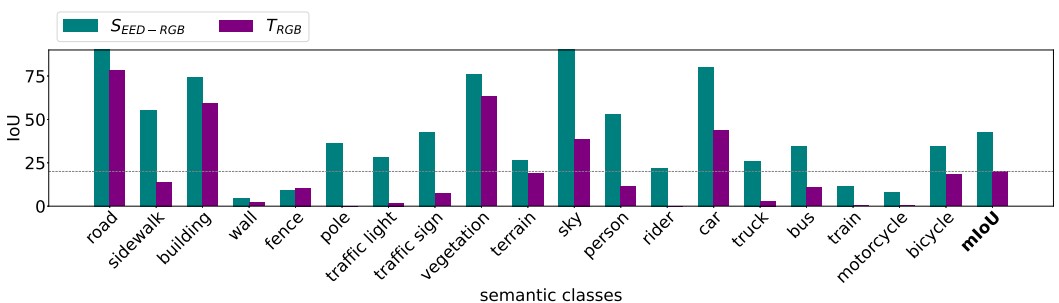

Figure 3: Class-specific cue influence for CNN based $S_{\text{EED-RGB}}$ and $T_{\text{RGB}}$ on Cityscapes.

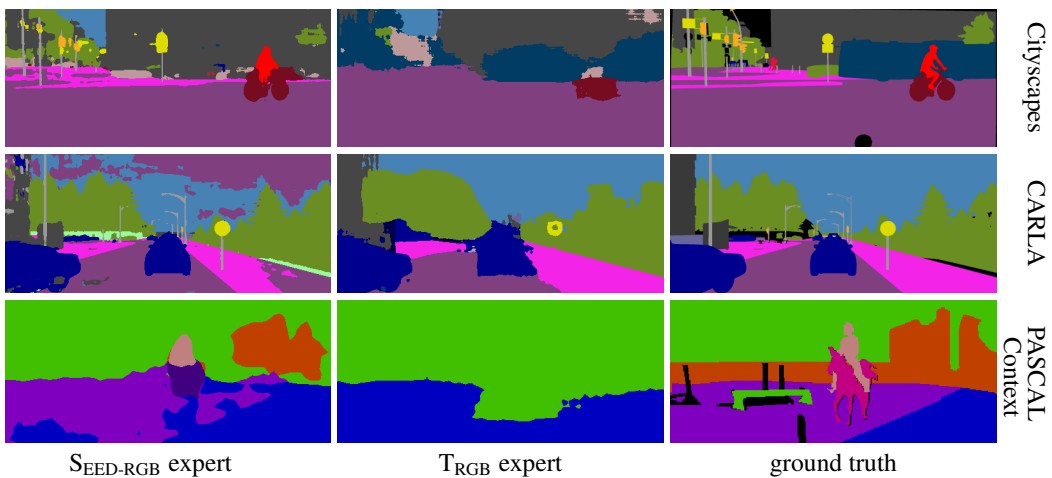

$S_{\text{EED-RGB}}$ expert        $T_{\text{RGB}}$ expert        ground truth

Figure 4: Comparison of the prediction of the two experts $S_{\text{EED-RGB}}$ (left) and $T_{\text{RGB}}$ (mid) for Cityscapes, CARLA and PASCAL Context. As a reference the ground truth is displayed in the third column (right).

we found that an IoU of at least $20\%$ for a given class starts to support the claim that the chosen expert extracts information for the given class. Hence, we see in fig. 3 that the S+C expert can deal with most of the classes while the T+C expert specializes to classes that usually cover large areas of the image, like vegetation, road and building although the texture expert was trained on a uniform class distribution. The results show that CNNs extract more discriminative information from colored shape than colored texture in real-world semantic segmentation tasks. Figures 10 to 11 show that the results reported on Cityscapes generalize to the other base datasets as well as to the transformer model. For visual examples see also figs. 4 and 16.

**Cue Influence Dependent on Location in an Image.** In this paragraph, we provide for all three datasets a detailed comparison of the same two experts from the previous paragraph, the shape expert based on EED having access to the cues S and C, and the texture expert based on the Voronoi images having access to the cues T and C. Here, cues are considered based on their location in an image.

We already noted in the previously presented class-specific study that the T+C expert focuses on classes covering larger areas of an image. Furthermore, we see a size dependence within a single class which is studied in fig. 5 for CARLA as base dataset for the classes road and person, where the former frequently occurs ($30\%$) and the latter is comparably rare ($2\%$) in terms of pixel counts. To study this effect in-depth, we trained a late fusion that processes the softmax outputs of both experts T+C and S+C, learning a pixel-wise weighting of both experts' outputs. We base our measurements on the output of the fusion model to decide for each pixel which cue contributes most on solving the learning task. By consistently adopting the prediction of the most influential expert, we calculate for each expert the segment-wise recall, which is the fraction of pixels in the ground truth segment covered by predictions of the same class. This metric measures the proportion of influence each

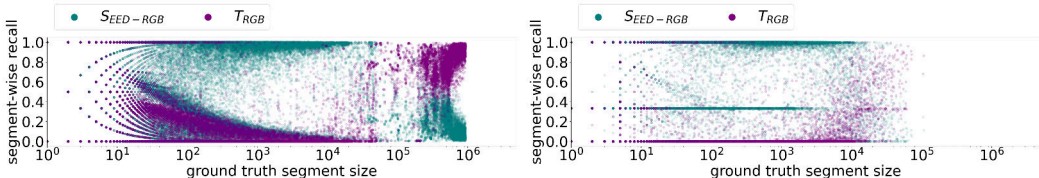

Figure 5: Coverage of experts over the classes 'road' (left) and 'person' (right) on the CARLA dataset. The recall on the y-axis is defined by the fraction of pixels in a ground-truth segment covered by a prediction of the same class.

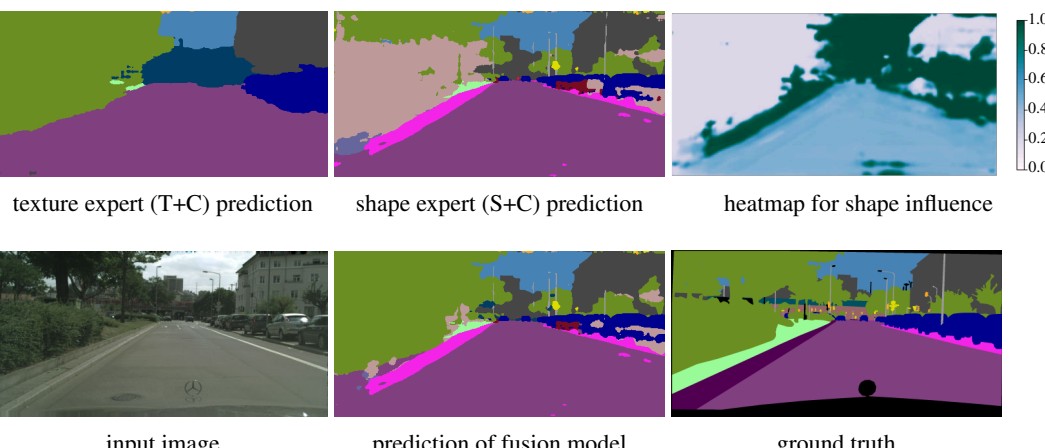

texture expert (T+C) prediction     shape expert (S+C) prediction     heatmap for shape influence

input image     prediction of fusion model     ground truth

Figure 6: A visual example of the predictions of an S+C and a T+C expert as well as the fusion model of an Cityscapes image. This is complemented with a heatmap depicting the pixel-wise input-dependent weighting for shape. Consequently, in lighter areas the texture cue is dominating.

expert has on a correct prediction of the specific segment. It can be seen that for the large road segments, the texture expert achieves a high segment-wise recall, indeed finding most of the road segments, while the shape expert has only a low segment-wise recall for those large segments. A similar trend can be observed for the rare class *person*.

In addition, we analyzed the influence dependent on the location in an image by our late fusion approach. A visual example for Cityscapes is provided in fig. 6. In general, we observe that the shape influences the fusion in regions containing boundaries of class-segments while priority is given to the texture inside larger segments. As can also be seen in fig. 6, the texture experts often have difficulties to accurately segment the boundaries of areas corresponding to a semantic class. We measured this effect quantitatively and provide results in table 4. The results reveal that the shape experts clearly outperform the texture expert on segment boundaries in terms of accuracy averaged over all boundary pixels. Herein, a pixel is considered as part of a segment boundary, if its neighborhood of four pixels distance (in Manhattan metric) contains a pixel of a different class according to the ground truth. On the segments' interior, on Cityscapes and PASCAL Context, the shape expert is on average still more useful than the texture expert. This is the other way round in CARLA. This can be explained by the observations that Cityscapes is in general relatively poor in texture and the textures in both real-world datasets are not very discriminatory. However, in the driving simulator CARLA, the limited number of different textures rendered onto objects is highly discriminatory and increases the texture expert's performance.

Comparing the pixel-wise predictions of two experts offers insights into ambiguities across different cues. If the texture cue expert predicts the class *car* but the shape expert predicts the class *road*, this can be a valuable source of redundancy and provide interpretable hints towards the safety of the overall prediction. The contradiction between two experts gives rise to an uncertainty metric. Quantitative and additional qualitative results for the late fusion of cue experts as well as qualitative examples for a contradiction / uncertainty heatmap are provided in the appendix.

Table 4: Comparison of the pixel accuracy for $S_{\text{EED-RGB}}$ and $T_{\text{RGB}}$ with respect to segment boundary pixels and segment interior pixels.

| pixel-averaged accuracy (%, ↑) | Cityscapes | | CARLA | | PASCAL Context | |
|---|---|---|---|---|---|---|
| | $S_{\text{EED-RGB}}$ | $T_{\text{RGB}}$ | $S_{\text{EED-RGB}}$ | $T_{\text{RGB}}$ | $S_{\text{EED-RGB}}$ | $T_{\text{RGB}}$ |
| segment interior | 88.59 | 75.10 | 82.63 | 89.83 | 56.48 | 39.96 |
| segment boundary | 56.49 | 37.16 | 70.44 | 47.94 | 38.83 | 25.99 |
| overall | 86.17 | 72.24 | 81.84 | 87.09 | 55.15 | 38.91 |

**Cue Influence in Different Architectures.** In addition to the previously discussed CNN-based cue experts, we also studied the cue extraction capabilities of a segmentation transformer, namely SegFormer-B1 Xie et al. (2021). The results on Cityscapes are provided in table 2. Although the CNN and transformer mIoU are almost on par when all cues are present, we observe a distinct increase in mIoU for the individual cue experts when using a transformer instead of a CNN. This holds for all T experts and S experts and their combinations with V and HS, respectively, with one exception which is the HED cue extraction. We expect that the HED mIoU suffers from the strong domain shift between HED images and original images, to which the HED expert is applied in table 2. Similarly, SegFormer achieves an mIoU of $54.81\% \pm 0.36$ pp. when applying HED as pre-processing. Nonetheless, qualitatively, i.e., in terms of the rankings of the different cue experts, we do not observe any serious differences between CNNs and transformers although pre-trained transformers are said to be more biased towards shape than CNNs Tuli et al. (2021). This indicates that the presence of a shape bias in semantic segmentation networks does not imply that transformers are less effective at learning from texture. These findings generalize to the class level where we observe an increase in performance in nearly all classes independent of the expert, but qualitatively the influence of the cues does not change. We conjecture that the increased cue performance of the transformer model results from the increased cross-domain performance as shown for Vision Transformers Yang et al. (2023) and for semantic segmentation transformers Wang et al. (2023).

## 5 CONCLUSION AND OUTLOOK

In this paper, we provided the first study on what can be learned by semantic segmentation DNNs from different image cues. Here, as opposed to image classification, studying cue influence is much more intricate and yields more specific and more fine-grained results. We introduced a generic procedure to extract cue-specific datasets from a given semantic segmentation dataset and studied several cue (combination) expert models, CNNs and transformers, across three different datasets and different evaluation granularities. We compared different cue (combination) experts in terms of mIoU on the whole dataset, in terms of semantic classes and in terms of image-location dependence. Our study provides the first empirical evidence for the following widely presumed statements: Except for grayscale images, there is no reduced cue combination that achieves a performance close to all cues. Cues influence the learning of DNNs over image locations, indeed shape often matters more in the vicinity of semantic class boundaries. Shape cues are important for all classes, texture cues mostly for classes where the corresponding segment covers large regions of the images in the dataset. Despite the difference in architecture, these findings generalize to transformers. Our generic cue extraction procedure (or at least parts of it) can also be utilized for studying biases in pre-trained off-the-shelf DNNs in cue conflict schemes, typical for that field of research. In addition, our data decompositions offer insights into ambiguities across different cues providing interpretable hints towards the safety of the overall prediction. For the future, a quantitative in-depth study is planned. Additionally, future research includes exploring different learning tasks like panoptic segmentation and different sensors like hyperspectral or infrared cameras as well as investigating how our analysis can be used to quantify the complexity of a learning task. We propose that the required cues to achieve a certain target accuracy could indicate learning complexity.

ACKNOWLEDGMENTS

TBA.

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

# A APPENDIX

In the appendix, we provide additional results as well as technical details of the procedure.

## A.1 CUE DECOMPOSITION

In fig. 7 we show exemplary images of cues extracted from the Cityscapes dataset. For all but the images containing the T cue, the cues can be extracted solely based on the base image (all cues). In contrast, the texture image is generated based on multiple images of the base dataset.

## A.2 TECHNICAL AND IMPLEMENTATION DETAILS

**Texture Data Generation Details.** A detailed scheme of the texture extraction procedure is given in fig. 8. The class-wise texture extraction is realized by masking all segments of one class and isolating each segment with the help of the border following algorithm proposed by Suzuki & Abe (1985). We discard segments with less than 36 pixels. The remaining segments are used to mask the original image and cut out the image patch of the enclosing bounding box of the segment. To enlarge the resulting patch pool, we apply horizontal flipping, random center crop and shift-scale-rotation augmentation. To mitigate the class imbalance in terms of pixel counts, we add more transformed patches for underrepresented classes than for classes which cover large areas of an image. It is essential to ensure that the transformations do not alter the texture. Once all the texture patches from all images within the underlying dataset have been extracted, they are randomly composed into mosaic images. We iteratively fill images of the same size as images from the base dataset with overlapping texture patches until we obtain completely filled mosaic images. To further reduce an unintended dependency on the shape cue we fill the mosaics in the original segmentation mask but assign each pixel to the same class (contour filled texture images). This implies that the segment boundaries are not discriminatory anymore. Based on a pool of contour filled texture images of different classes we can create a surrogate segmentation task by filling the segments of arbitrary segmentation masks with the generated texture images. We choose Voronoi diagrams Torquato (2002) as surrogate segmentation task and fill each cell randomly but uniformly with respect to the class with crops of the pool of contour filled texture images. To generate an entire dataset, we create and fill as many Voronoi diagrams as the number of images in the base dataset. Note, that this extraction method does not allow for one to one correspondence between a base dataset image and a texture image.

**HED Implementation Details.** To generate object defining edge maps we use the implementation and pre-trained model of Harary et al. (2022) which bases on the PyTorch re-implementation of the original method by Niklaus (2018).

**EED Data Generation Details.** EED is an anisotropic diffusion technique based on Partial Differential Equations (PDEs) Perona et al. (1994); Weickert et al. (1998). Starting with the original image as the initial value, it utilizes a spatially dependent variation of the Laplace operator to propagate color information along edges but not across them, thus leading to texture being largely removed from images and higher level features being largely preserved. For the production of the EED data Weickert et al. (1998) we have used a variation proposed in Heinert et al. (2024) that avoids circular artifacts and preserves shapes particularly well by applying spatial orientation smoothing as proposed for Coherence Enhancing Diffusion in Weickert (1999). The PDE is solved using explicit Euler, channel coupling from (Weickert & Welk, 2006, p. 321) and the discretization described in (Weickert et al., 2013, p. 380-391).

The diffusion parameters are the same for all three data sets, Cityscapes, CARLA and PASCAL Context:

- Contrast parameter $\lambda = 1/15$.
- Gaussian blurring kernel size $k = 5$ and standard deviation $\sigma = \sqrt{5}$.
- Time step length $\tau = 0.2$ and artificial spatial distance $h = 1$.
- Number of time steps $N_{EED} = 8192$.

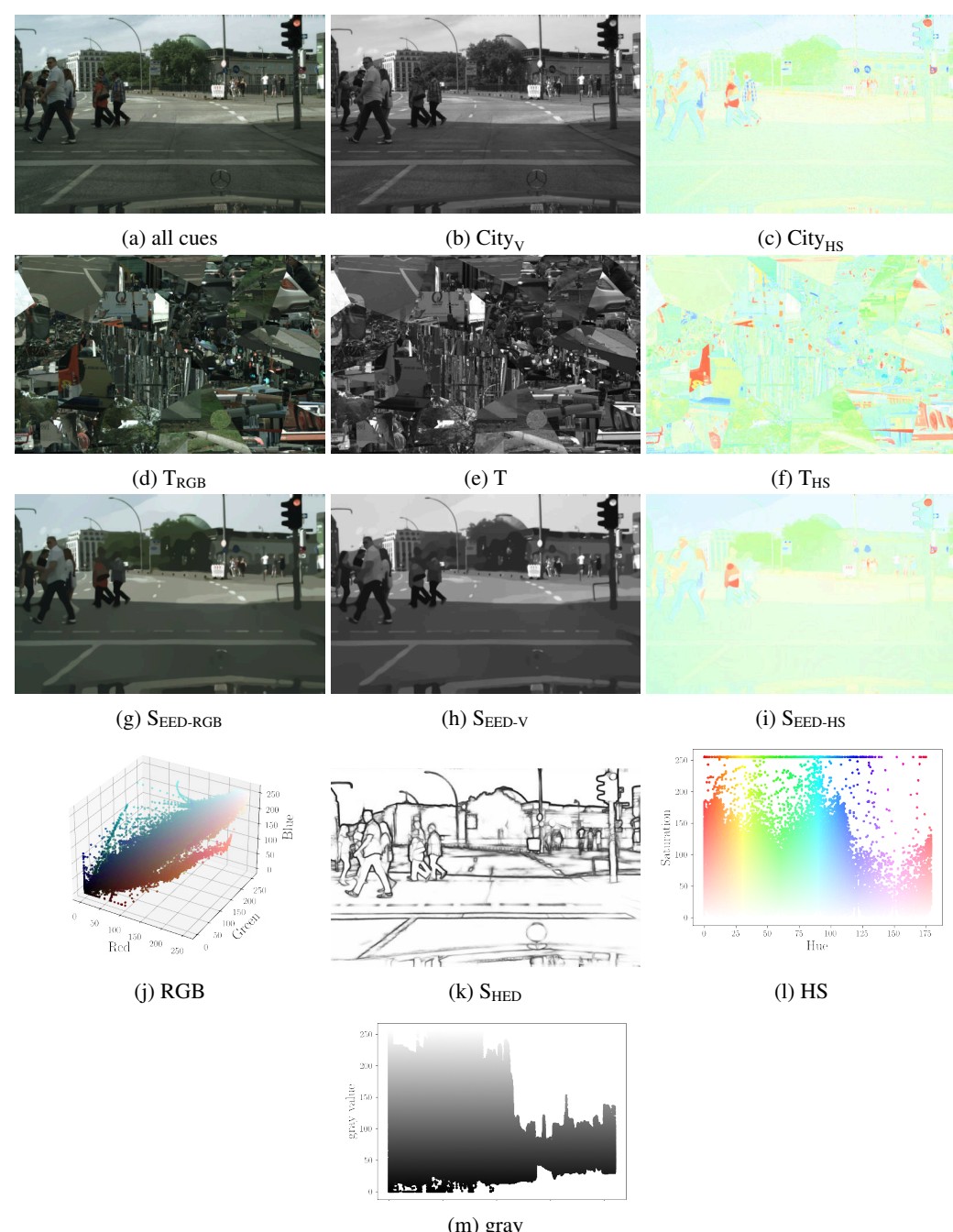

Figure 7: Overview of the cue decomposition of a Cityscapes image (all cues) into texture, shape, hue and saturation (HS) and gray components. For the color cues the RGB, HS and V value distribution of the image are scattered for visualization purpose.

- Discretization parameters $\alpha = 0.49$ and $\beta = 0$.

For a visual example see fig. 7.

**Carla Data Generation Details.** Although CARLA claims to support online texture switching as of version 0.9.14, this feature is limited to specific instances and found to be inadequate for the purposes of this study. As a consequence, we manually modified each material instance, replacing

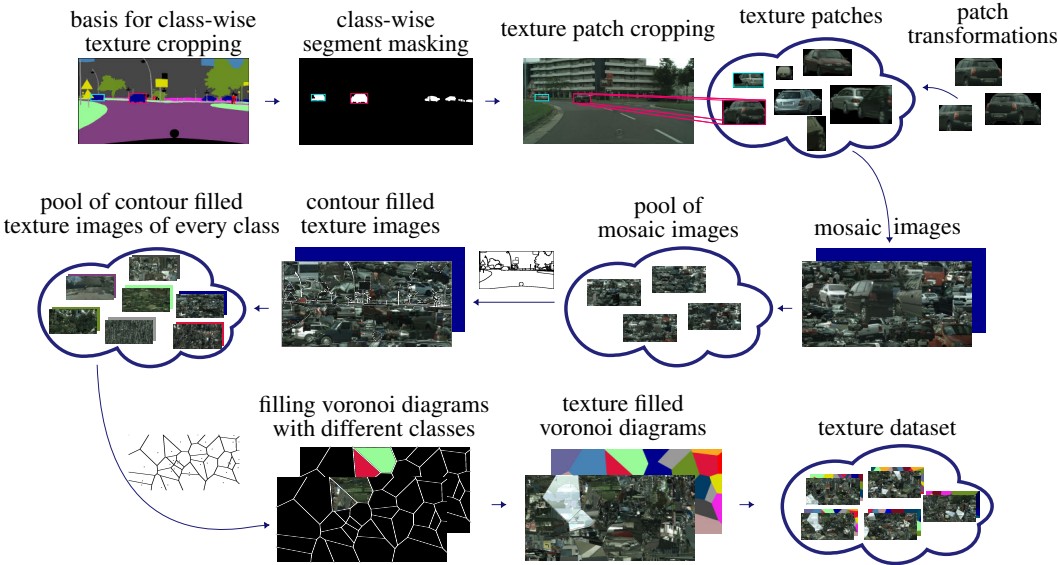

Figure 8: Detailed scheme of the texture cue extraction process. It consists of the three main steps: class-wise patch extraction, class-wise mosaic image construction and segmentation dataset creation based on Voronoi diagrams.

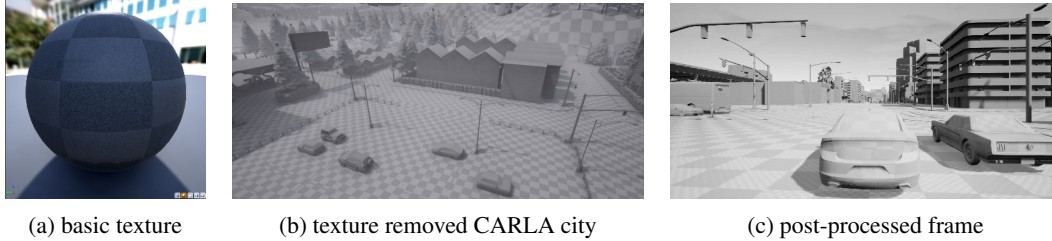

(a) basic texture      (b) texture removed CARLA city      (c) post-processed frame

Figure 9: Basic components to extract shape with grayness via texture removal in CARLA.

surface textures with a basic default texture pattern (gray checkerboard). Since the sky is not a meshed object, it was not possible to manipulate its texture. Instead, we set the weather conditions to clear noon to achieve a uniform texture. The dataset is recorded by a vehicle in CARLA with an RGB camera and semantic segmentation sensor in the ego perspective driving in autopilot mode as described in section 4.1. In a post-processing step the images are gray scaled to remove the sky color. The basic components of the procedure are visualized in fig. 9.

**Model Implementation Details.** For the CNN-based models we adapt the DeepLabV3 model from torchvision[2] to our needs. For the transformer models we use the MMSegmentation framework MMSegmentation Contributors (2020).

## A.3 ADDITIONAL NUMERICAL EXPERIMENTS

**Comparison of Cue Influence Without Domain Shift.** The texture and shape cue experts face a domain shift when evaluated on the corresponding base dataset image. Except for the texture extraction procedure all cue extraction methods can be applied as an online transformation during inference. This allows us to compare the cue influence without domain shift for the shape experts. We observe that the S cue based on object contours ($S_{HED}$) mostly outperforms other experts trained on shape cues, also when the latter receive additional color cue information, see table 5. The generation of the texture cue dataset does not allow a one to one correspondence between a texture and

---

[2]`https://github.com/pytorch/vision/blob/main/torchvision/models/segmentation/deeplabv3.py`

Table 5: Cue influence in terms of mIoU performance for the CNN experts when evaluated in-domain, i.e., the validation dataset is pre-processed with the same cue extraction method as the training dataset. Each column is sorted in ascending order according to the in-domain cue performance on Cityscapes.

| Cityscapes | | CARLA | | | PASCAL Context | | |
|---|---|---|---|---|---|---|---|
| | mIoU | | mIoU (%, ↑) | rank change w.r.t. Cityscapes CNN | | mIoU (%, ↑) | rank change w.r.t. Cityscapes CNN |
| no info | $0.25 \pm 0.35$ | no info | $0.38 \pm 0.44$ | → | no info | $0.11 \pm 0.11$ | → |
| V | $6.39 \pm 0.04$ | V | $6.01 \pm 0.08$ | → | V | $2.27 \pm 0.04$ | → |
| HS | $9.33 \pm 0.18$ | HS | $14.88 \pm 0.38$ | → | HS | $3.35 \pm 0.10$ | → |
| RGB | $11.31 \pm 0.52$ | RGB | $15.77 \pm 0.57$ | → | RGB | $4.91 \pm 0.05$ | → |
| $S_{EED-V}$ | $45.02 \pm 0.51$ | $S_{EED-V}$ | $60.73 \pm 2.20$ | → | $T_{HS}$ | $25.80 \pm 0.10$ | ↗7 |
| | | $S_{textureless}$ | $61.45 \pm 1.70$ | | | | |
| $S_{EED-HS}$ | $46.00 \pm 0.46$ | $S_{EED-HS}$ | $62.20 \pm 1.89$ | → | $S_{EED-HS}$ | $29.13 \pm 1.15$ | → |
| $S_{EED-RGB}$ | $48.47 \pm 0.45$ | $S_{HED}$ | $62.65 \pm 1.88$ | ↗1 | $S_{EED-V}$ | $33.20 \pm 0.62$ | ↘2 |
| $S_{HED}$ | $55.80 \pm 0.59$ | $S_{EED-RGB}$ | $65.83 \pm 0.63$ | ↘1 | $S_{EED-RGB}$ | $35.33 \pm 0.78$ | ↘1 |
| $City_{HS}$ | $59.89 \pm 0.74$ | $CARLA_{HS}$ | $70.34 \pm 1.56$ | → | $Pascal_{HS}$ | $36.10 \pm 0.34$ | ↗1 |
| $City_{V}$ | $64.21 \pm 0.60$ | $CARLA_{V}$ | $73.17 \pm 5.19$ | → | $S_{HED}$ | $37.63 \pm 0.15$ | ↘2 |
| all cues | $65.22 \pm 0.47$ | all cues | $75.71 \pm 1.50$ | → | $T_{V}$ | $39.37 \pm 0.50$ | ↗3 |
| $T_{HS}$ | $79.63 \pm 2.22$ | $T_{HS}$ | $94.30 \pm 0.77$ | → | $T_{RGB}$ | $39.73 \pm 0.55$ | ↗1 |
| $T_{RGB}$ | $81.20 \pm 1.34$ | $T_{RGB}$ | $97.08 \pm 1.12$ | → | $Pascal_{V}$ | $45.39 \pm 0.71$ | ↘3 |
| $T_{V}$ | $86.20 \pm 1.43$ | $T_{V}$ | $97.53 \pm 0.13$ | → | all cues | $45.45 \pm 0.18$ | ↘3 |

base dataset image. However, we can evaluate the expert performance on a texture dataset generated from the validation images of the base dataset. For Cityscapes and CARLA, we find that the T cue outperforms all other cues. We conclude that learning texture is easier but either suffers stronger from the domain shift or overfits to its training domain.

**Cue Influence on Frequency-weighted Segmentation Performance.** Dependent on the use-case, different metrics may be better suited to quantify the performance of a model. MIoU is often used in semantic segmentation, if all classes are equally important. The frequency-weighted Intersection over Union (fwIoU) can be used to weight each class importance depending on their frequency of appearance Ulku & Akagündüz (2022). Comparing the ranking of the cue influences measured by mIoU (see tables 2 and 3) and by fwIoU (see table 6), we observe only minor differences for real-world datasets. For the CARLA dataset we see a slightly higher influence of the texture which aligns with our findings that the T cue is mostly valuable for larger segments.

Table 6: Cue influence w.r.t. frequency-weighted segmentation performance.

| Cityscapes CNN | | | CARLA | | | PASCAL Context | | |
|---|---|---|---|---|---|---|---|---|
| | CNN fwIoU (%, ↑) | rank change w.r.t. mIoU | | CNN fwIoU (%, ↑) | rank change w.r.t. mIoU | | fwIoU (%, ↑) | rank change w.r.t. mIoU |
| V | $25.78 \pm 0.148324$ | → | V | $17.70 \pm 0.69$ | → | V | $06.50 \pm 0.10$ | → |
| HS | $33.22 \pm 0.4711688$ | → | $S_{HED}$ | $26.07 \pm 1.65$ | → | $S_{HED}$ | $07.97 \pm 1.64$ | ↗1 |
| RGB | $40.06 \pm 1.2136721$ | → | HS | $45.10 \pm 1.19$ | → | HS | $09.07 \pm 0.23$ | ↘1 |
| $S_{HED}$ | $45.62 \pm 5.6935929$ | → | $S_{EED-V}$ | $46.00 \pm 3.91$ | ↗1 | RGB | $12.67 \pm 0.12$ | → |
| $S_{EED-HS}$ | $51.04 \pm 5.0604348$ | ↗1 | RGB | $46.83 \pm 1.25$ | ↘1 | $T_{HS}$ | $17.30 \pm 0.20$ | → |
| $T_{V}$ | $57.32 \pm 1.7512852$ | ↘1 | $S_{EED-HS}$ | $63.97 \pm 4.70$ | → | $T_{V}$ | $23.50 \pm 1.04$ | ↗2 |
| $T_{RGB}$ | $58.90 \pm 3.4907019$ | → | $T_{V}$ | $70.83 \pm 4.15$ | → | $T_{RGB}$ | $23.67 \pm 2.23$ | ↘1 |
| $T_{HS}$ | $59.04 \pm 2.9896488$ | → | $S_{EED-RGB}$ | $71.48 \pm 1.18$ | ↗2 | $S_{EED-HS}$ | $26.10 \pm 1.84$ | ↘1 |
| $S_{EED-V}$ | $62.54 \pm 6.1313131$ | → | $T_{RGB}$ | $78.35 \pm 1.60$ | → | $S_{EED-V}$ | $32.73 \pm 2.36$ | → |
| $S_{EED-RGB}$ | $78.14 \pm 2.9441467$ | → | $T_{HS}$ | $78.83 \pm 1.03$ | ↘1 | $S_{EED-RGB}$ | $39.97 \pm 0.75$ | → |
| $City_{HS}$ | $88.26 \pm 0.181659$ | → | all cues | $82.18 \pm 2.36$ | ↗2 | $PASCAL_{HS}$ | $44.33 \pm 0.32$ | → |
| $City_{V}$ | $89.24 \pm 0.181659$ | → | $CARLA_{V}$ | $86.33 \pm 3.43$ | → | $PASCAL_{V}$ | $52.40 \pm 0.61$ | → |
| all cues | $89.84 \pm 0.1516575$ | → | $CARLA_{HS}$ | $88.43 \pm 5.34$ | ↘2 | $PASCAL_{RGB}$ | $53.50 \pm 0.00$ | → |

**Cue Influence on Different Semantic Classes.** Figures 10 to 11 show that the results reported on Cityscapes generalize to the other base datasets as well as to the transformer model. However, for the CARLA base dataset we observe that the texture is more discriminatory compared to the real-world datasets (cf. column 4 'rank change' in table 3) and the performance of the $T_{RGB}$ expert on classes like road, sidewalk and building surpasses the performance of the comparable shape expert $S_{EED-RGB}$. In fig. 16 we provide additional qualitative results for this finding.

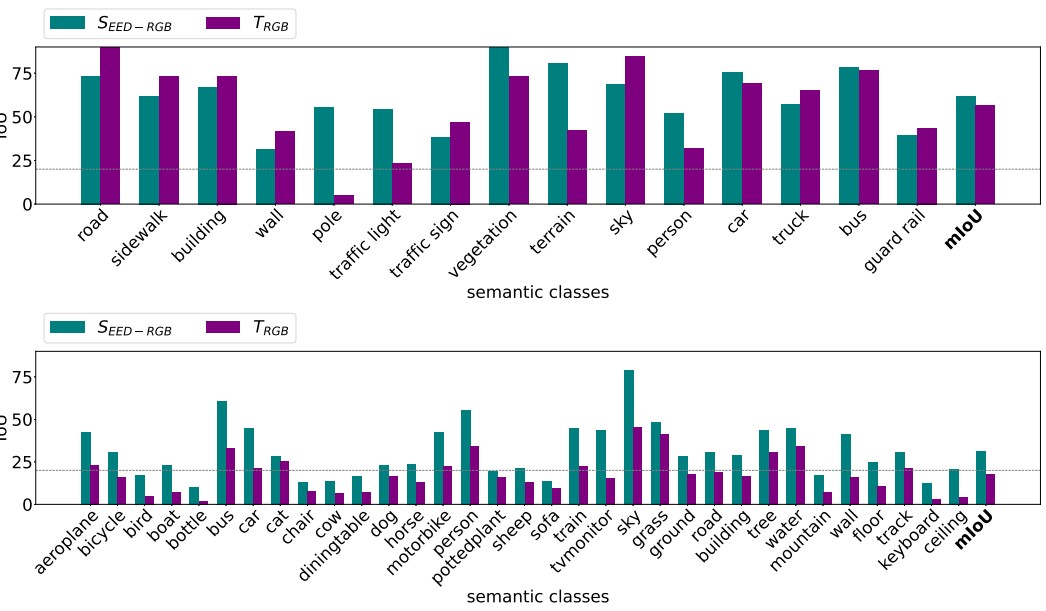

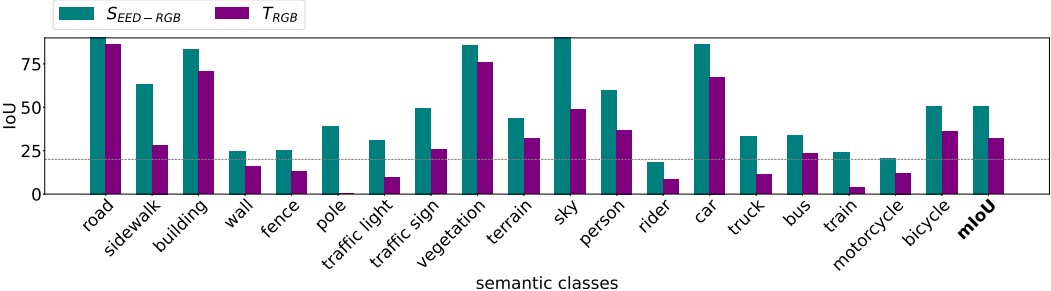

Figure 10: Class-specific cue influence for the shape and texture cue, both with RGB color cue, on CARLA and PASCAL Context.

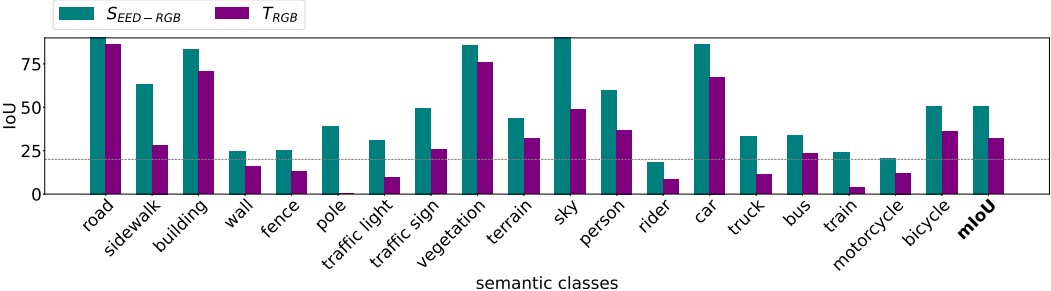

Figure 11: Class-specific cue influence for the transformer based shape and texture cue, both with RGB color cue, on Cityscapes.

**Cue Influence Dependent on Location in an Image.** In this section we provide additional numerical results on the study of the two experts $S_{\text{EED-RGB}}$ and $T_{\text{RGB}}$ on pixel level. In our experiments for the CARLA base dataset on pixel level we found that the S+C and T+C cues are complementary. Fusing $S_{\text{EED-RGB}}$ and $T_{\text{RGB}}$ improves the overall scene understanding by a notable margin. The fusion model achieves a performance of $78.10\%$ mIoU on the CARLA test set which is about $19$ pp. superior to the test set performance of $S_{\text{EED-RGB}}$ and more than $22$ pp. superior to when relying only on $T_{\text{RGB}}$ (cf. fig. 10). In CARLA the described dominated shape influence on segment boundaries (cf. table 4) is more pronounced and thus exploited by the fusion model, see fig. 12. Additional qualitative results of the fusion prediction and the corresponding pixel-wise weighting of the expert influence is depicted in fig. 13.

A qualitative overview of the predictions of all cue experts is shown in figs. 17 to 19 for Cityscapes, CARLA and PASCAL Context, respectively. We observed for the street scene base datasets that already a non-negligible amount of information, like the position of a car, can be learned solely based on the color value of a pixel (cf. figs. 17 and 18 last row).

Reliable predictions are of major concern when using DNNs in safety critical applications like autonomous driving or medical imaging. We suggest that cue experts with a similar performance serve as different sources of evidence for a prediction. If the texture cue expert predicts a car but the shape expert predicts road, this can be a valuable source of redundancy and provide interpretable hints

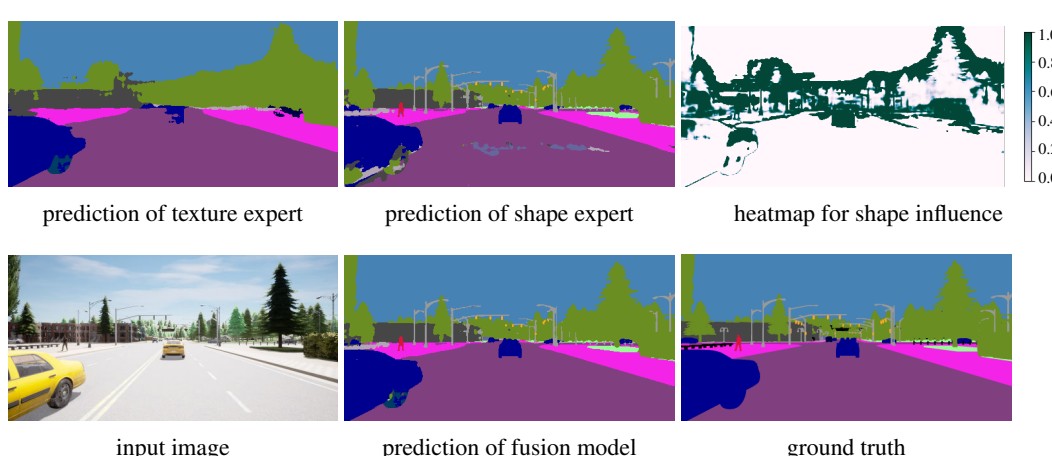

Figure 12: A comparison of the prediction of the fusion model and the $T_{RGB}$ and $S_{EED-RGB}$ experts on a CARLA test set image. The heatmap shows where and how much each expert's prediction influences the prediction of the fusion model. Dark green pixels mean the fusion model bases its prediction on the shape expert, while light purple pixels mean it is based on the texture expert.

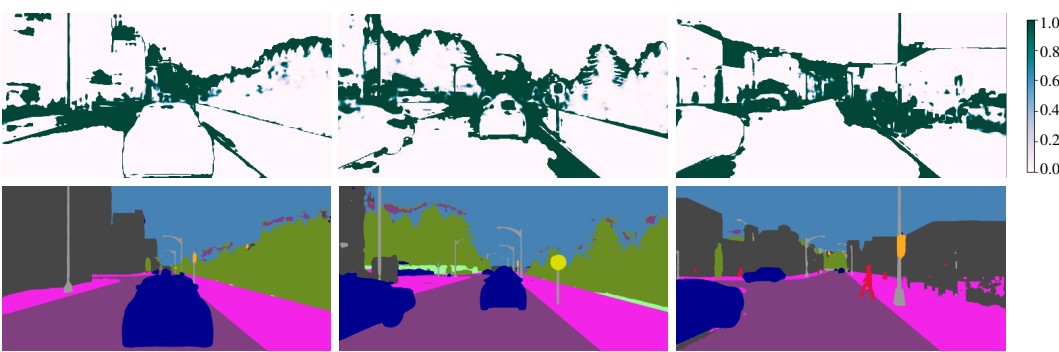

Figure 13: Fusion of the softmax prediction of $S_{EED-RGB}$ and $T_{RGB}$ on CARLA (see fig. 16 for the single expert prediction). The heatmap shows how much influence the expert's prediction has in the fusion model. Dark green pixels mean the fusion model bases its prediction on the shape expert, while light purple pixels mean it is based on the texture expert.

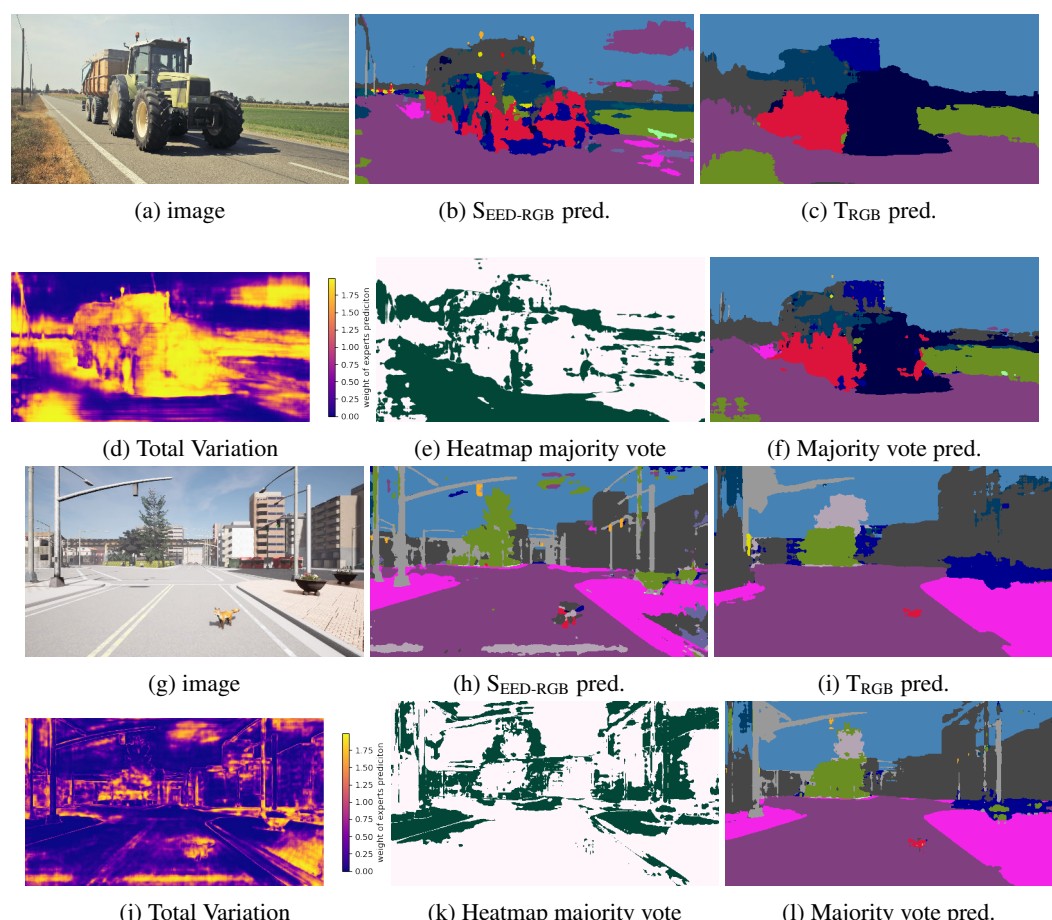

(a) image      (b) $S_{EED\text{-}RGB}$ pred.      (c) $T_{RGB}$ pred.

(d) Total Variation      (e) Heatmap majority vote      (f) Majority vote pred.

(g) image      (h) $S_{EED\text{-}RGB}$ pred.      (i) $T_{RGB}$ pred.

(j) Total Variation      (k) Heatmap majority vote      (l) Majority vote pred.

Figure 14: Example of a contradiction heatmap based on the total variation distance for the predicted class distributions of the $S_{EED\text{-}RGB}$ and $T_{RGB}$ expert on Cityscapes and CARLA for an unusual road user. In d) light denote a high total variation distance and can be understood as high prediction uncertainties. In e) pixels in green correspond to predictions based on the shape expert.

towards the safety of the overall prediction. This information could also be weighted with respect to the findings of our study, e.g., that we rely more on the shape expert for segment boundary pixels. We generate an uncertainty heatmap by calculating the total variation distance $||p_S - p_T||_1$ between the predicted class distributions given by the softmax activations $p_S$ and $p_T$ of the respective experts, $S_{EED\text{-}RGB}$ and $T_{RGB}$. In each pixel the joint prediction of the two experts is the prediction of the expert which is more confident. That is, for each pixel the prediction is set to the class with the highest softmax activation among both experts. We qualitatively evaluate our approach on street scenes with unusual road users in the real and synthetic world. The results are visualized in fig. 14. In both cases we observe a high total variation distance, indicating that the experts' predictions contradict on the unusual road user. This can be used as an uncertainty metric for the joint prediction. We plan to explore this direction in future work.

**Cue Influence in Different Architectures.** The transformer experiments conducted on the PASCAL Context dataset (see table 7) demonstrate results comparable to those obtained on the Cityscapes dataset (cf. table 2). These findings suggest that the influence of cues remains largely consistent between CNN and transformer architectures, as the order of cues has not experienced significant changes (see last column of table 7 for reference).

To gain deeper insights, we investigate, whether the backbone depth of the CNN expert models has an impact on the cue extraction capability. We analyze the influence of the number of ResNet layers for the experts $S_{EED\text{-}RGB}$, $T_{RGB}$ and all cues. We trim the ResNet backbone to either 2, 3 or

Table 7: Cue influence in terms of mIoU performance drop on PASCAL Context for DeepLabV3 with ResNet backbone and SegFormer. Cue description follows the listing in table 1. MIoU gaps to maximal performance are stated in percent points (pp.). "PASCAL" refers to PASCAL Context.

| | S | T | Color V | Color HS | CNN mIoU (%, ↑) | CNN gap (pp., ↓) | transformer mIoU (%, ↑) | transformer gap (pp., ↓) | change in rank w.r.t. CNN |
|---|---|---|---|---|---|---|---|---|---|
| no info | | | | | $0.11 \pm 0.11$ | 45.34 | $0.45 \pm 0.34$ | 43.14 | → |
| V | | | ✓ | | $2.27 \pm 0.04$ | 43.18 | | | |
| HS | | | | ✓ | $3.35 \pm 0.10$ | 42.10 | | | |
| $S_{HED}$ | ✓ | | | | $4.71 \pm 1.15$ | 40.74 | $5.04 \pm 0.47$ | 38.55 | ↘[1] |
| RGB | | | ✓ | ✓ | $4.91 \pm 0.05$ | 40.54 | | | |
| $T_{HS}$ | | ✓ | | ✓ | $11.39 \pm 0.36$ | 34.06 | $13.70 \pm 0.14$ | 29.89 | → |
| $T_{RGB}$ | | ✓ | ✓ | ✓ | $17.75 \pm 0.82$ | 27.70 | $19.99 \pm 0.21$ | 23.60 | → |
| $S_{EED-HS}$ | ✓ | | | ✓ | $17.80 \pm 1.67$ | 27.65 | $21.15 \pm 0.49$ | 22.44 | ↘[1] |
| $T_V$ | | ✓ | ✓ | | $18.43 \pm 0.47$ | 27.02 | $20.27 \pm 0.23$ | 23.32 | ↗[1] |
| $S_{EED-V}$ | ✓ | | ✓ | | $25.80 \pm 2.40$ | 19.65 | $22.78 \pm 0.65$ | 20.81 | → |
| $S_{EED-RGB}$ | ✓ | | ✓ | ✓ | $31.32 \pm 0.82$ | 14.13 | $32.09 \pm 0.75$ | 11.50 | → |
| $PASCAL_{HS}$ | ✓ | ✓ | | ✓ | $36.10 \pm 0.34$ | 9.35 | $35.03 \pm 0.16$ | 8.56 | → |
| $PASCAL_V$ | ✓ | ✓ | ✓ | | $45.39 \pm 0.71$ | 0.06 | $43.22 \pm 0.04$ | 0.37 | → |
| all cues | ✓ | ✓ | ✓ | ✓ | $45.45 \pm 0.18$ | 0.00 | $43.59 \pm 0.36$ | 0.00 | → |

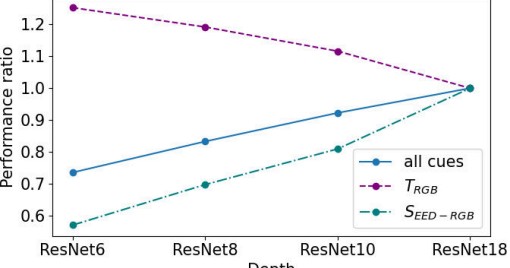

| layer width | # layers | mIoU | # parameters |
|---|---|---|---|
| 256 | 2 | 11.31 | 84,288 |
| 256 | 3 | 11.79 | 150,336 |
| 256 | 5 | 11.95 | 282,432 |
| 128 | 30 | 11.06 | 487,584 |
| 256 | 14 | 10.49 | 876,864 |
| 512 | 5 | 11.55 | 1,121,920 |
| 512 | 7 | 12.03 | 1,648,256 |

Figure 15: Layer study: Left: Change in performance w.r.t. the number of ResNet layers. The performance is normalized w.r.t. the mIoU obtained by the corresponding cue expert with a ResNet18 backbone evaluated on the base dataset. Right: Comparison of the performance for the color expert with respect to the model capacity. The backbone of the FCN with only $1 \times 1$-convolutions is varied between 2 to 30 layers with varying width.

4 ResNet layers each with a single ResNet block, and term them ResNet6, ResNet8 and ResNet10, respectively. When training each ResNet on the cue-specific dataset and evaluating it on the base dataset, we observe that the texture expert improves performance with fewer ResNet layers. In contrast, the performance of the shape expert and of the expert trained on the original data with all cues increases with more layers. The results suggest, that relevant features are learned in earlier layers since a moderate depth is enough to correctly predict segments. Furthermore, we observe that the texture expert overfits to its training domain for deeper ResNet architectures, leading to a decreased performance on the original dataset. In contrast, the shape expert, presumably needs a larger field of view to predict segments based on shape features since a deeper architecture improves the performance.

Additionally, we compare the performance of the color expert with respect to different FCN backbones varying in the width and the number of layers. The results in fig. 15 show that increasing the capacity up to an factor of 19.5 (last row) does not significantly increase the model performance. Our evaluation is limited to 1,648,256 parameters due to reaching the maximal GPU RAM capacity of an A100 with 80 GB RAM. We conclude that the model capacity in terms of learnable parameters used in our experiments of the main manuscript (first row) is not a limiting factor for the segmentation performance.

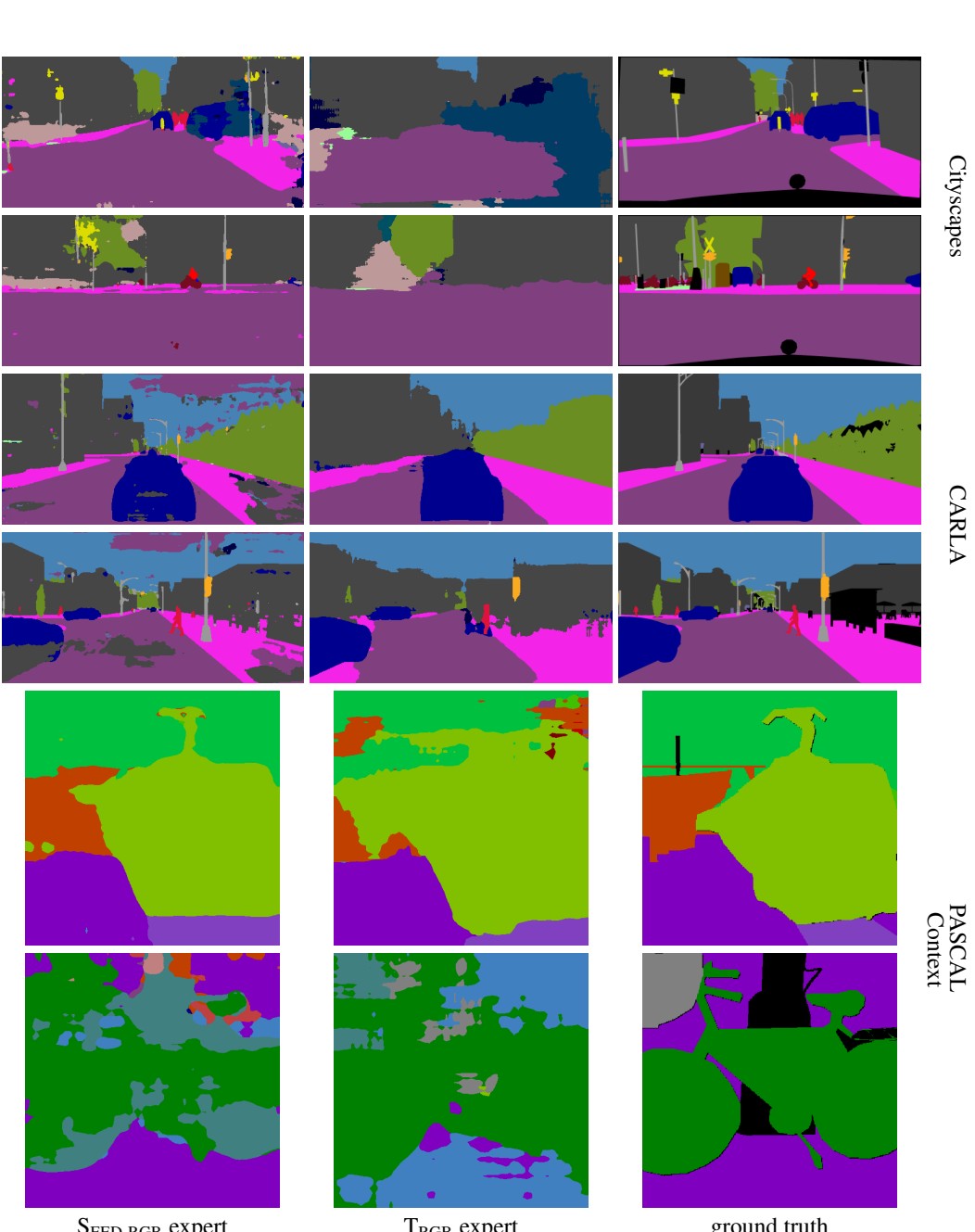

Figure 16: Comparison of the prediction of the two experts $S_{EED-RGB}$ (left) and $T_{RGB}$ (mid) for Cityscapes, CARLA and PASCAL Context. As a reference the ground truth is displayed in the third column (right).

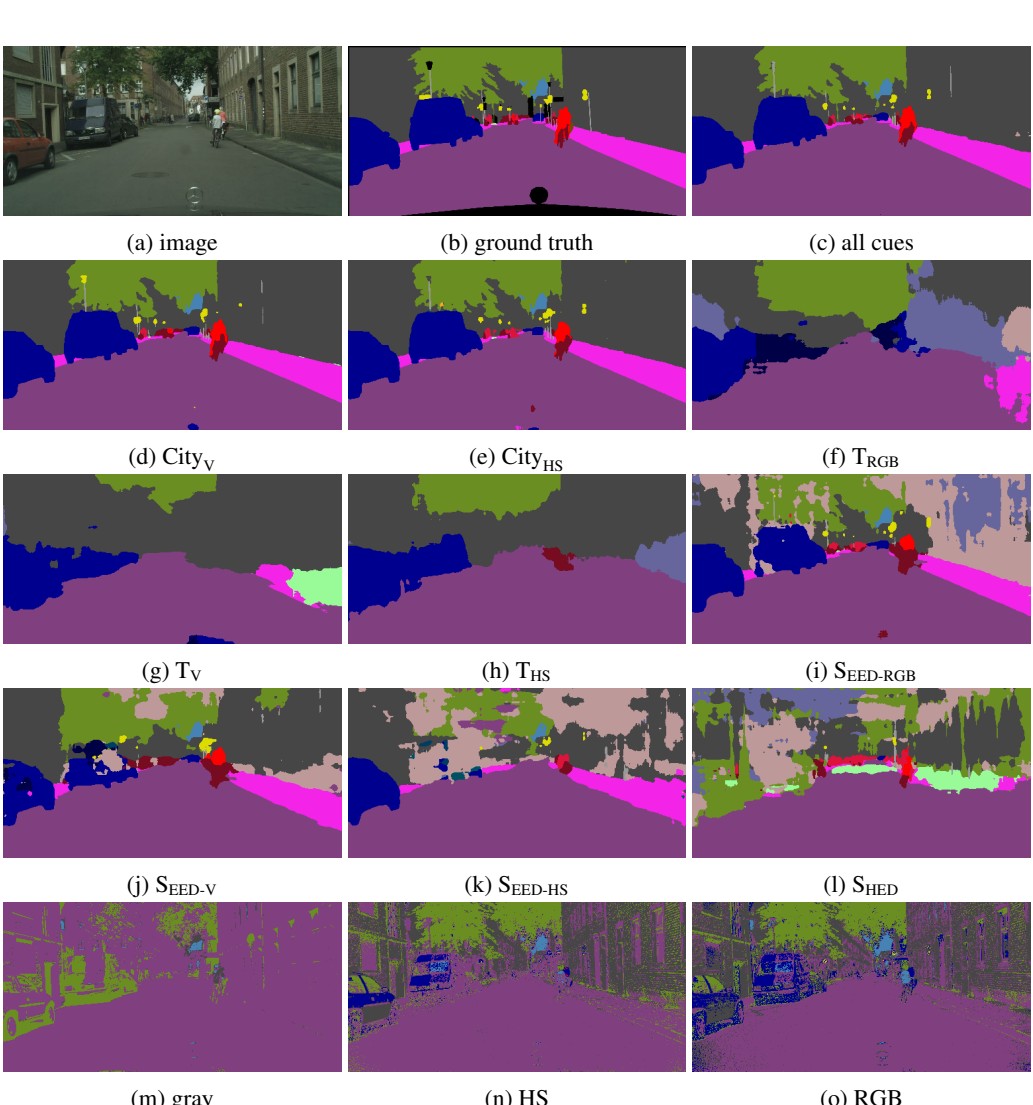

(a) image  (b) ground truth  (c) all cues

(d) City$_V$  (e) City$_{HS}$  (f) T$_{RGB}$

(g) T$_V$  (h) T$_{HS}$  (i) S$_{EED\text{-}RGB}$

(j) S$_{EED\text{-}V}$  (k) S$_{EED\text{-}HS}$  (l) S$_{HED}$

(m) gray  (n) HS  (o) RGB

Figure 17: Overview of the predictions of all cues for a Cityscapes image

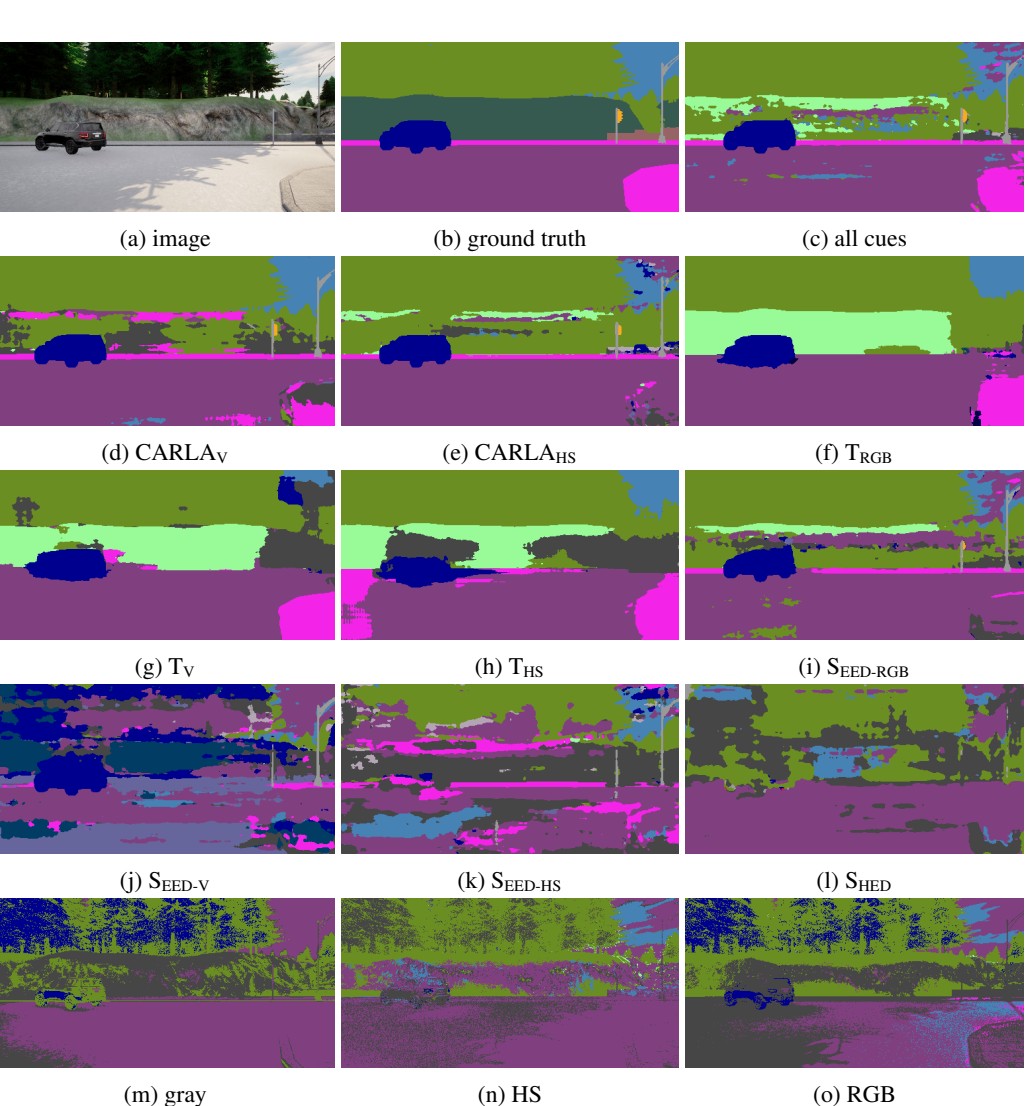

(a) image  (b) ground truth  (c) all cues

(d) CARLA$_V$  (e) CARLA$_{HS}$  (f) T$_{RGB}$

(g) T$_V$  (h) T$_{HS}$  (i) S$_{EED-RGB}$

(j) S$_{EED-V}$  (k) S$_{EED-HS}$  (l) S$_{HED}$

(m) gray  (n) HS  (o) RGB

Figure 18: Overview of the predictions of all cues for a CARLA image

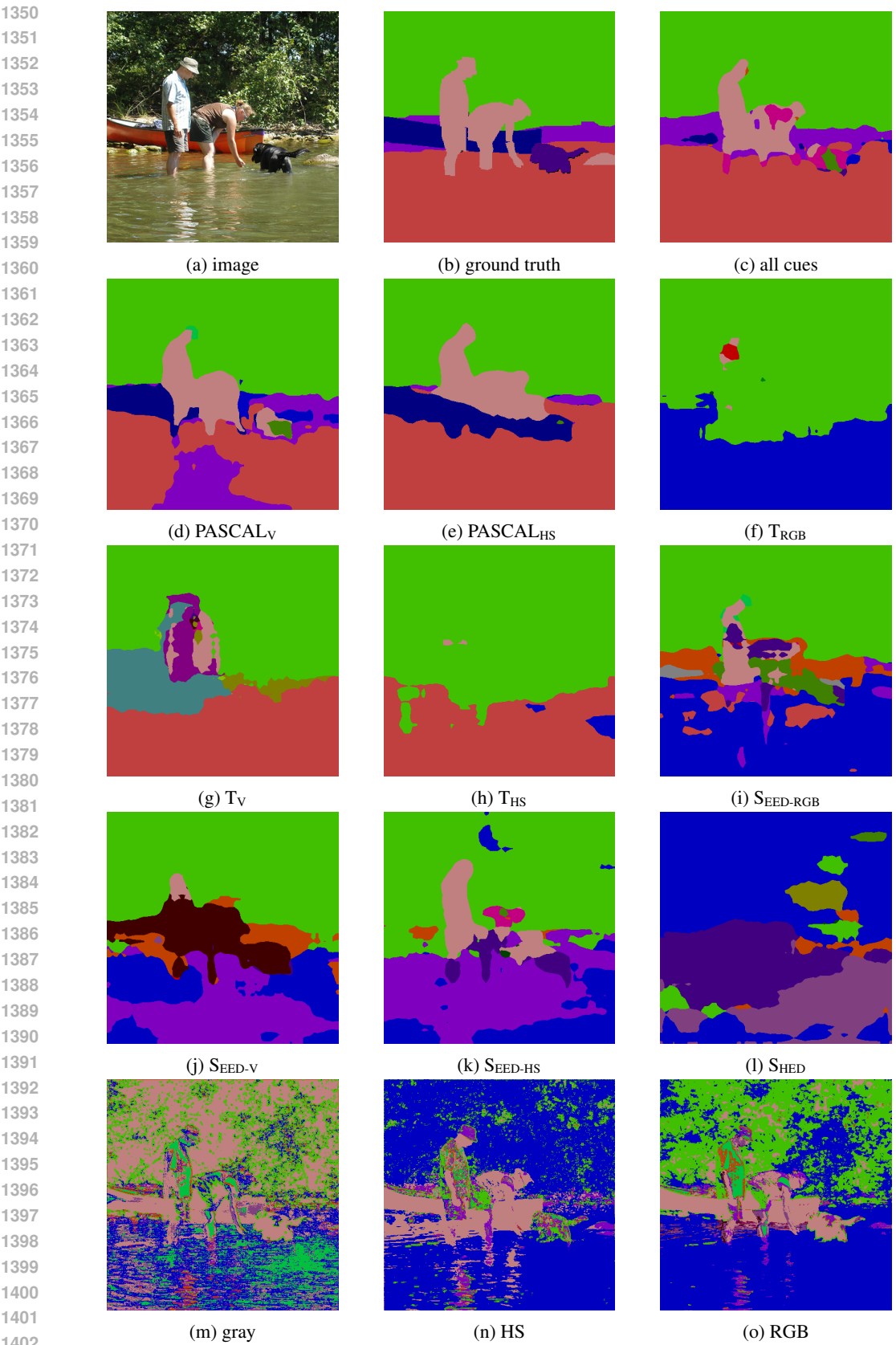

Figure 19: Overview of the predictions of all cues for a PASCAL Context image.

