# OpenReview forum: "On the Influence of Shape, Texture and Color for Learning Semantic Segmentation"
_ICLR.cc/2025/Conference — Submitted to ICLR 2025_

### Official Review · Reviewer_j3xz · 2024-10-24

**Soundness:** 2
**Presentation:** 3
**Contribution:** 2
**Rating:** 5
**Confidence:** 4

**Summary:**

In this work, the authors try to answer an important question, which is what can DNNs learn from image cues such as shape, texture, and color. They argue that this is the opposite of what is typically done in previous literature, as previous works rather focus on understanding how much a trained neural network relies on different cues. In short, the aim of this paper is to understand what are the important cues for successful trainings.
To answer this question, the authors propose to decompose a given dataset into multiple different sub-datasets, with each one including one or more cues to learn from. They set up different experiments (cue-experts) from different combinations of cues in the context of semantic segmentation and using both CNNs and Transformers.
The outcome of these experiments is that there is no clear winner between relying on shapes rather than texture is essential for good performance, but rather a combination of shape and colors.

**Strengths:**

1. In my opinion, this paper addresses a very interesting problem, namely, which are the cues to learn from. A thorough analysis could be beneficial for other problems such as explainability, Domain Adaptation, and  Domain Generalization.

2. The paper is well-written and clear, and I particularly appreciate Table 1 summarising all the possible settings.

3. A good number of experiments across different datasets and architectures have been conducted.

**Weaknesses:**

1. My main concern about this paper is that the results are unclear. The rank change column is particularly helpful in understanding the current scenario, and all datasets clearly have different cue orders when taken individually across datasets. I understand that the authors claim that they do not drastically differ, but still, they are not consistent in drawing clear conclusions.

2. The claimed results are not surprising, and I do not see a clear impact or the direction of this paper. For example, it is obvious to me that cues such as texture is overall better than color since texture also contains color information, as also stated by the authors. Furthermore, is not surprising that using all cues is better.

3. One final major concern is about the validity of some experiments. In particular, on how evaluation is done for the color cue expert. To my understanding, in such cases, all convolutions have been replaced with 1x1 convolution kernel. These produce very poor results, but we do not know whether this is because of a model that does not have enough capacity or the color is not a useful cue.
Furthermore, I am not sure that it is a valid method to train a model one cue and then test it on the original images. In some cases, this could make sense, but in other cases, the distribution shift between source and target images could be so high that the model is not able to perform well.

**Questions:**

1. When exacting the cue colors, is 1x1 convolution used for all layers or only for the first? In the former case, it is really hard to understand whether the network has enough capacity to learn a difficult task such as semantic segmentation, while the latter case would invalidate the experiment as the following layers would have a larger receptive field.

2. The second part of the sentence at L193: “As it preserves color, it extracts the cues S+V+HS, and analogously to the treatment of the C cue provides the cues S+V as well.” is not clear to me.

3. About weaknesses 3, I wonder whether it is possible to test each cue expert under the scenario it has been trained on, and not on the original images. Maybe the authors could clarify this point and motivate this choice.

---

> ### Author Response · Authors · 2024-11-24
> **Official Comment by Authors (1/2)**
>
> First, we would like to thank the reviewer for the constructive and positive feedback. We sincerely appreciate the recognition of the importance of the problem we address with "a good number of experiments across different datasets and architectures".
> In the following, we discuss the raised concerns which helped us refine the paper. All adaptations we made are highlighted in violet in the revised paper.
>
> **[W1] unclear results when looking at the rank change**
>
> *Response:*
> Learning semantic segmentation is a difficult task which is still not fully understood. In our paper, we propose a method which gives insight into the cue influence down to pixel level for the first time.
> However, the unique characteristics of each dataset result in differences in how much information the cues contribute to the learning task.
> The rank change column is supposed to abstract from the numeric results that suffer from variance. Rank changes (except for the synthetic dataset) mainly occur when the numeric results are very close to each other and often can be explained by the variance. For the synthetic data the rank changes are more pronounced due to the rendering characteristics (see also our response to [W2, Q2] of reviewer pTje).
> Nonetheless, it is noteworthy that a consistent and intuitive but not yet proven trend emerged across multiple datasets which ranks color, texture and shape cues and texture-shape combinations. Furthermore, we calculated the rank correlation across different datasets. The Pearson correlation coefficient of 0.934 for the Cityscapes and the CARLA cue ranking denotes a high correlation. We adapted the formulation in paragraph "Cue Influence on Mean Segmentation Performance" in the paper to break down the arguments more precisely.
>
>
> **[W2] The claimed results are not surprising, and I do not see a clear impact or the direction of this paper. Furthermore, is not surprising that using all cues is better.**
>
> *Response:*
> We acknowledge that the findings may appear intuitive; however, we would like to highlight that this is the first study in semantic segmentation to experimentally validate these intuitions across diverse depth levels, down to the pixel level and across different architectures. We believe this contribution is particularly valuable, as also noted by reviewer o1oW, since intuitions can sometimes prove incorrect or only partially true, as seen with the shape bias hypothesis for CNNs in classification (see Geirhos et al.).
> The same study shows, using all cues of ImageNet at the same time can lead to a bias. If a certain bias is helpful in solving the task it would be intuitive that this cue encodes more information than other cues. However, we find it interesting, that from an information retrieval perspective neither shape nor texture alone clearly dominates the learning success.
> From our perspective, proving these not surprising aspects has an impact on the deeper understanding of the learning process of DNNs for semantic segmentation. We would like to also draw attention to the aspect that we propose a general method which allows to decompose any custom dataset. This allows not only to analyze the specific cue influences but also to use the decomposed data to, e.g., study biases of pre-trained DNNs.
>
> **[W3] and [Q3] One final major concern is about the validity of some experiments. In particular, on how evaluation is done for the color cue expert. [Is the capacity or the usefulness of the cue the reason for the poor performance?]
> Furthermore, I am not sure that it is a valid method to train a model one cue and then test it on the original images. [How about testing] each cue expert under the scenario it has been trained on?**
>
> *Response:*
> We appreciate the detailed review and clarify the concerns with respect to 1) the color cue expert and 2) the cue evaluation method.
> 1) For the color cue expert we investigated FCN models with only 1x1-convolutions with different capacity to balance between over-parametrization, GPU RAM (80 GB) constraints and comparability.
> All models achieved a comparable performance why we at first refrained from including this into the paper. We will add the results in the appendix to demonstrate that the poor performance is not due to the model capacity.
> 2) We agree with the reviewer, that the domain shift in the evaluation method is worth investigating.
> For the shape cues this evaluation is reasonable since the transformation is per image and can be applied before processing the image by the cue expert. The main findings of this study have been described in the first paragraph of 4.2. 'Numerical Results' and the detailed table has only been presented in the appendix (cf. 'Comparison of Cue Influence Without Domain Shift') due to the page limitation. For the revised manuscript, we enlarged the table and included the domain-shift-free evaluation for the texture cue (combinations) as well.

---

> > ### Comment · Reviewer_j3xz · 2024-11-26
> >
> > Thank you for the detailed response. After thoroughly reviewing your comments and considering the concerns raised by other reviewers, I remain convinced that the paper lacks sufficient depth in analysis and results. While the experiments provide some high-level insights, I believe they are not enough for a publication. As such, I have decided to maintain my current score.

---

> ### Author Response · Authors · 2024-11-24
> **Official Comment by Authors (2/2)**
>
> **[Q1] When exacting the cue colors, is 1x1 convolution used for all layers or only for the first? In the former case, it is really hard to understand whether the network has enough capacity to learn a difficult task such as semantic segmentation, while the latter case would invalidate the experiment as the following layers would have a larger receptive field.**
>
> *Response:*
> We restrict the color cue model to 1x1-convolutions for the mentioned reason, to not invalidate the experiments by an enlarged field of view. As discussed for [W3] we investigated models with different capacities to mitigate the addressed problem.
>
> **[Q2] The second part of the sentence at L193 [...] is not clear to me.**
>
> *Response:*
> We revised the sentence to improve readability.
>
> **References:**
> - Robert Geirhos et al. "ImageNet-trained CNNs are biased towards texture; increasing shape bias improves accuracy and robustness." ICLR, 2018.

---

### Official Review · Reviewer_o1oW · 2024-10-29

**Soundness:** 3
**Presentation:** 3
**Contribution:** 3
**Rating:** 6
**Confidence:** 3

**Summary:**

The paper investigates the importance of different image cues (e.g. color, shape, texture) for successful training of deep semantic segmentation models. Different from previous work, this study does not focus on the analysis of conventionally pretrained models. Instead, it is based on "expert" models trained on custom datasets containing only one or a combination of selected cues. To enable this type of analysis, the paper proposes several approaches to transform original image datasets into versions that contain only specific cues. The analysis includes two real image datasets: Pascal Context and Cityscapes, as well as a single synthetic dataset based on CARLA which provides more control over the image generation process. Additionally, the study explores considers convolutional as well as transformer based architectures.

**Strengths:**

Previous shape and textures bias studies rarely considered dense prediction tasks, so this looks like a valuable contribution.

The proposed study is the first that investigates the effect of individual cues (and different combinations of cues) on the training process of the semantic segmentation models.

The paper consolidates a method for extraction of different image cues from natural images. This enables transformation of the original datasets into variants that contain a single image cue or selected combination of cues.

**Weaknesses:**

Presentation quality could be improved.
I suggest placing the tables and figures right after being referenced in the text.

I found the Texture Cue Extraction paragraph confusing. The main manuscript should include more details and be more descriptive. I am not sure how the Voronoi diagrams are created and do they depend on the content of the corresponding image. Are class frequencies and distributions preserved in this texture cue dataset?

The last paragraph in 4.1 should also describe the evaluation protocol. What kind of input each expert model considers during the evaluation? This is partly addressed at the end of the first paragraph in 4.2. However, I think it deserves a separate paragraph to clarify the edge cases, so that the reader has no uncertanties when considering the numerical results. I would also consider including the EED and HED results with processing in the corresponding table.

I am not entirely convinced with the conclusions of the discussion about the influence of the size on the performance of texture and shape experts. The texture expert might specialize in these classes not because the size of their segments, but just because they are more frequent, or their texture is more unique and therefore easier to discriminate. I am not sure. It obviously depends on the properties of the training dataset. I think this requires further considerations.

I feel like the paper lacks some practical takeaways or ideas for future work that would use the findings of this study for improving the performance or robustness of deep models for semantic segmentation.

**Questions:**

Do you expect similar conclusions for panoptic segmentation as well? This might open some new questions. For example, if the same clues have equal effect on the performance in thing and stuff classes? Mask transformers [1] disentangle the panoptic inference into mask localization and classification. It would be interesting to see which clues are more important for localization, and which ones for classification. What are your thoughts about this and the future work?

In table 2, the convolutional model and the transformer achieve similar performance when considering regular images with all cues. However, the transformer consistently outperforms the convolutional model by a large margin on datasets which consider only subsets of clues. What is the reason behind that?

---

> ### Author Response · Authors · 2024-11-24
> **Official Comment by Authors (1/2)**
>
> We thank the reviewer for their constructive and positive feedback. We sincerely appreciate the recognition of the novelty of our study and method as well as the detailed review and thoughtful questions, which helped us to identify open questions and improve our manuscript. Below, we summarize and address the comments individually. All adaptations we made are highlighted in violet in the revised paper.
>
> **[W1] Presentation quality could be improved [by] placing the tables and figures right after being referenced.**
>
> *Response:* We rearranged some figure and table positions with respect to the layout feedback. We hope the adaptions meet the reviewer's expectation.
>
> **[W2] The main manuscript should include more details and be more descriptive [w.r.t. the Texture Cue Extraction]. How [are] the Voronoi diagrams created? Do they depend on the content of the corresponding image? Are class frequencies and distributions preserved in this texture cue dataset?**
>
> *Response:* We thank the reviewer for the constructive and precise feedback. To improve the comprehensibility of our texture extraction method, we added more details addressing the questions of the reviewer in the main manuscript in section 3 Texture (T) Cue Extraction as well as in the detailed explanation in the Appendix.
> We provide here a short answer to the last two questions.
> The Voronoi diagrams explicitly do not depend on the semantic contours of the corresponding image. In fact, the Voronoi diagrams serve as surrogate segmentation task to remove the original shapes in the scene. However, they could be replaced by any other polygonal partitioning. The texture patches, which are filled into the individual Voronoi cells, are gathered over multiple images to ensure a certain diversity and size. We explicitly refrained from preserving the class distribution to not bias the cue influence on class level. For reference, we trained Voronoi diagrams which are filled according to the base dataset class distribution. We kindly refer to our answer to [Q1] of reviewer bPD6 for more details.
>
> **[W3] [...] The evaluation protocol is partly addressed at the end of the first paragraph in 4.2. However, I think it deserves a separate paragraph to clarify the edge cases. What kind of input each expert model considers during the evaluation?
> (I would also consider including the EED and HED results with processing in the corresponding table.)**
>
> *Response:*
> We agree with the reviewer that our paper can be improved by inserting a dedicated paragraph for the evaluation protocol. We appreciate this constructive feedback and expanded the manuscript in section 4.1. To not confuse the reader by switching between evaluation methods we exclude online transformation for data evaluation in tab. 2 and 3. Instead, we included a table to the appendix with all in-domain performances, i.e., the validation dataset is pre-processed with the same cue extraction method as the
> training dataset.
>
> **[W4] I am not entirely convinced with the conclusions of the discussion about the influence of the size on the performance of texture and shape experts. The texture expert might specialize in these classes [...] because they are more frequent, or their texture is more unique and therefore easier to discriminate. [...] I think this requires further considerations.**
>
> *Response:*
> Our analysis of the cue influence on different semantic classes revealed that the combination of T+C cues is not helpful to learn all classes but those which often cover large segments of the image and therefore frequently occur during training (cf. Fig.3). We appreciate the detailed review which brings into question the same aspects as we thought of: are there specific reasons to focus on these classes?
> The review revealed that our analysis had potential for a more concise description, so we revised the paper according to the following aspects:
> Firstly, we describe more precisely that we generated the texture dataset with a uniform class distribution. To this end, the frequency of classes is nearly equal, and we conclude that the class frequency is not a main cause that texture experts specialize on a subset of classes.
> Secondly, we investigated if the shape expert (S_EED-RGB) or the texture expert (T_RGB) is more useful for learning semantic segmentation with respect to the segment size within one single class (see Fig. 6).
> For the CARLA dataset where the shape and texture expert perform similarly well, we found that large segments within the frequent class road have a high segment-wise recall only for the texture expert. The same trend is visible within the rarely occurring class person.

---

> ### Author Response · Authors · 2024-11-24
> **Official Comment by Authors (2/2)**
>
> **[W5] I feel like the paper lacks some practical takeaways or ideas for future work that would use the findings of this study for improving the performance or robustness of deep models for semantic segmentation.**
>
> *Response:*
> One of our motivations is to explore how DNNs perceive the world relates to contradiction of different sources of evidence. If the texture cue expert predicts a car but the shape expert predicts road, this can be a valuable source of redundancy and provide interpretable hints towards the safety of the overall prediction. To exploit this, a general understanding of the strengths and weaknesses of the various experts are needed which we have provided in the present study. Independent of the practical utility, our study contributes to the fundamental understanding of how DNNs learn from image data.
> Our data decomposition offers insights into ambiguities across different cues. This enables our approach to quantify the complexity involved in learning a dataset, helping to identify inherent ambiguities and uncertainties in the data. We outline this use-case in the outlook section and provide exemplary contradiction heatmaps for unusual road objects for Cityscapes and CARLA in the appendix.
>
> In addition, we think it is worth investigating how our analysis can be used to quantify the complexity of a learning task. We hypothesize that evaluating which and how many cues are needed to predict a specific class up to a certain accuracy can serve as a learning complexity quantity.
>
> To gain deeper insights, we conducted a layer study revealing that shallow DNNs seem sufficient to learn from texture data whereas a comparably deeper architecture can learn more from shape cues.
> Besides exploring different learning tasks, we believe that the cue influence of different sensors like hyperspectral or infrared cameras is interesting to explore. Furthermore, we also see additional value in the decomposed data for, e.g., evaluating pre-trained models w.r.t. texture or shape biases which is ongoing work.
>
> **[Q1]
> Do you expect similar conclusions for panoptic segmentation as well? This might open some new questions. For example, if the same clues have equal effect on the performance in thing and stuff classes? Mask transformers [1] disentangle the panoptic inference into mask localization and classification. It would be interesting to see which clues are more important for localization, and which ones for classification. What are your thoughts about this and the future work?**
>
> *Response:*
> This is a very interesting question which allows for new comparisons and insights. Based on our findings for semantic segmentation, that the cue influence is more dependent on the pixel location rather than the class, we
> hypothesize that a similar result might be observable for panoptic segmentation. However, since panoptic segmentation combines multiple learning tasks, it is worth investigating if the cue influence depends on the learning task by analyzing and comparing different model architectures including MaskFormer. From our perspective, both hypothesis are worth to investigate in more depth. We appreciate this idea and included it as future research in the manuscript.
>
>
> **[Q2] [...] the transformer consistently outperforms the convolutional model by a large margin on datasets which consider only subsets of clues. What is the reason?**
>
> *Response:*
> We conjecture that the increased cue performance of the transformer model results from the increased cross domain performance as shown for Vision Transformers (Tvt by Yang et al.) and for semantic segmentation transformers (CDAC by Wang et al.). We added this statement to the paper to address this question.
>
>
> **References:**
> - Wang, Kaihong, et al. "CDAC: Cross-domain attention consistency in transformer for domain adaptive semantic segmentation." Proceedings of the IEEE/CVF International Conference on Computer Vision. 2023.
> - Yang, Jinyu, et al. "Tvt: Transferable vision transformer for unsupervised domain adaptation." Proceedings of the IEEE/CVF Winter Conference on Applications of Computer Vision. 2023.

---

> ### Comment · Reviewer_o1oW · 2024-11-26
>
> Thank you for your response and clarifications! After considering the authors' responses and feedback from other reviewers, I decided to maintain my score. As others have also noted, the study's insights are limited and lack practical takeaways, which is the primary reason for not assigning a higher score.

---

### Official Review · Reviewer_pTje · 2024-11-02

**Soundness:** 3
**Presentation:** 3
**Contribution:** 2
**Rating:** 6
**Confidence:** 3

**Summary:**

This work studies the impact of shape and texture on training DNN. They develop a generic procedure to decompose a given dataset into multiple ones, each of them only containing either a single cue or a chosen mixture. they develop a generic procedure to decompose a given dataset into multiple ones, each of them only containing either a single cue or a chosen mixture. The study on three datasets reveals that neither texture nor shape clearly dominates the learning success, however, a combination of shape and color but without texture achieves surprisingly strong results. The findings hold for convolutional and transformer backbones. In particular, qualitatively there is almost no difference in how both of the architecture types extract information from the different cues.

**Strengths:**

1. This paper comprehensively analyzes the impact of shape, texture, gray and their combination on semantic segmentation tasks, and provides a method to derive a texture-only dataset. This paper compares the different effects of these image cues on CNN and Transformer.
2. The structure of the paper is relatively clear and the introduction of the methods is relatively detailed.
3. Studying the impact of different image cues on semantic segmentation is very meaningful for designing networks and contributes to the deep learning community.

**Weaknesses:**

1. This paper shows the effects of different visual cues on semantic segmentation, but lacks a detailed analysis of why these effects occur, and explores which modules and operations are introduced into the model to reduce these effects.
2. In Table 3, we find that the rank change of the CARLA dataset is large relative to the Cityscapes dataset. Please provide more explanations why different performances are shown on different datasets.
3. Figure 6 is not clear enough.

**Questions:**

1. After we have studied the impact of different visual cues on the segmentation effect of the model, can we use certain deep learning operations to specifically improve the effect of the model?
2. In Table 3, we find that the ranking changes of the CARLA dataset are larger than those of the Cityscapes dataset. Why do visual cues show such differences on different datasets?

---

> ### Author Response · Authors · 2024-11-24
> **Official Comment by Authors**
>
> First, we would like to thank the reviewer for the constructive and positive feedback, which acknowledges the comprehensiveness of our study and its contribution to the deep learning community. In the following, we comment on the stated weaknesses and how we address them. At the end we answer the reviewer's questions. All adaptations we made are highlighted in violet in the revised paper.
>
> **[W1] and [Q1] This paper shows the effects of different visual cues on semantic segmentation, but lacks a detailed analysis of why these effects occur, and explores which modules and operations are introduced into the model to reduce these effects.
> After we have studied the impact of different visual cues on the segmentation effect of the model, can we use certain deep learning operations to specifically improve the effect of the model?**
>
> *Response:* One of our primary goals is to demonstrate that the way DNNs perceive the world can be broken down into distinct sources of evidence. In our study, we analyze the influences of the inherent cues in an image and address the questions "What can be learned from individual cues? How much does each cue influence the learning success? And what are the synergy effects between different cues?"
> The weakness and question stated by the reviewer addresses a somewhat different perspective, namely an analysis from the model rather from the data perspective. This opens up diverse research opportunities which would constitute a standalone paper. For a deeper analysis of the cue influence on the learning success we conducted an additional experiment which relates to the aspect mentioned by the reviewer and was included into the manuscript. We investigated the influence of backbone depth on the performance of cue-expert-models. The experiment shows that shape experts profit from deeper neural networks. In contrast, for datasets with dominating texture features shallow networks might even improve the overall performance.
>
>
> **[W2] and [Q2] The rank change of the CARLA dataset is large relative to the Cityscapes dataset. Why do visual cues show such differences on different datasets?**
>
> *Response:* Although our results show a general trend in the cue influences across all datasets ("C experts are mostly dominated by T experts as
> well as S experts, and those are in turn dominated by S+T expert"), each dataset has its own characteristics. These characteristics influence the specific cues. The resulting rank changes are more pronounced in domains with greater distinctions, such as synthetic versus real-world data, but are still highly correlated (Pearson correlation coefficient of 0.934). Unlike Cityscapes and PASCAL Context, the CARLA dataset is synthetic, captured in a rendered city based on a limited amount of textures, assets and synthetic lighting conditions. This leads to less diverse but more discriminatory texture and shape which we discuss in "Cue Influence Dependent on Location in an Image".
>
>
> **[W3] Figure 6 is not clear enough.**
>
> Response: In figure 6, we visualize the cue influence w.r.t. the segment size within one class. We thank the reviewer for pointing out that this figure needs more clarification. We updated our analysis description accordingly in the paper in paragraph "Cue Influence Dependent on Location in an Image".
> The results are visualized for a frequently occurring class as well as a rare class in terms of pixel count. We investigated if the shape expert (S_EED-RGB) or the texture expert (T_RGB) is more useful for learning semantic segmentation with respect to the segment size within one single class.
> For the CARLA dataset, where the shape and texture expert perform similarly well, we found that large segments of the class road have a high segment-wise recall for the texture expert whereas the segment-wise recall of the shape expert drops for larger segments. A similar trend can be observed for the rare class "person". We conclude consistency independent of the occurrence frequency of the class.

---

> > ### Comment · Reviewer_pTje · 2024-11-26
> >
> > Thanks to the author for responding to the question, I will keep my vote score.

---

### Official Review · Reviewer_bPD6 · 2024-11-03

**Soundness:** 3
**Presentation:** 3
**Contribution:** 2
**Rating:** 5
**Confidence:** 4

**Summary:**

The paper presents an analysis of the influence of shape, texture, and color on semantic segmentation performance, proposing a methodology that leverages augmented datasets to isolate these specific ‘cues’ and assess their individual contributions. To achieve this, the paper uses HSV channels to represent colors, edge detection to represent shapes, and ground-truth mask cropping to isolate textures. Through extensive combinations of these cues, the authors evaluate the performance of semantic segmentation using both real-world datasets and the CARLA simulator, applying this methodology to both convolutional and transformer-based architectures.

**Strengths:**

The paper is easy to read, with a significant level of detail dedicated to the experiments and results. The study is comprehensive in its scope, providing extensive combinations of cue-based datasets and evaluating them across different model architectures.

By isolating and recombining shape, color, and texture cues, the paper sheds light on how each of these elements contributes to segmentation performance. This could be beneficial for understanding data domain gaps and predicting failure modes of segmentation models.

**Weaknesses:**

The study’s approach, while straightforward, lacks sufficient depth in its experimental design and interpretation. The experiments primarily provide surface-level insights without delving into a more profound analysis of underlying factors.

Most of the findings and corresponding discussion, for example, that shape cues correspond heavily to semantic boundaries and that textures play a significant role in broader regions—are valid but not particularly surprising and interesting to computer vision researchers. Most of the results presented in the paper fall into this category. The insights gained from this study feel somewhat limited, leaving the reader with few novel takeaways. As such, it may fall marginally below the acceptance threshold in its current form.

There are several technical and methodological concerns, please see Questions.

**Questions:**

Using Voronoi diagrams, assigned randomly to semantic classes, raises questions about whether the generated "texture" dataset truly reflects the original distribution of classes. How about assigning semantic classes to Voronoi diagrams by their frequency in the original dataset?

The reliance on an external edge detector for shape cues introduces external biases. For example, if the edge detectors are trained on semantic boundaries, using them will introduce additional semantic information. How about using low-level filter-based edge detectors?

When shape and color are combined (no texture), the input image becomes piecewise constant. Since the segmentation maps are also piecewise constant, this operation simplifies the mapping between input and output. The authors present this result as surprising, yet it is somewhat intuitive given the characteristics of segmentation tasks.

mIoU may not be a good metric to fully capture the performance across classes with different frequencies. A more robust analysis would include additional metrics such as frequency-weighted IoU and pixel accuracy.

Figures 10 and 11 are interesting and important. I suggest moving them to the main paper.

---

> ### Author Response · Authors · 2024-11-24
> **Official Comment by Authors (1/2)**
>
> First, we would like to thank the reviewer for the constructive and positive feedback, which recognizes the comprehensiveness and scope of our study which "sheds light on how [shape, color, and texture cues] contributes to segmentation performance's". Furthermore, we appreciate the recognition of its relevance to domain adaptation and model failure prediction.
> In the following we will address the stated weaknesses and answer the reviewer's questions individually. All adaptations we made are highlighted in violet in the revised paper.
>
> **[W1] "The study’s approach, while straightforward, lacks sufficient depth in its experimental design and interpretation[...]"**
>
> *Response:* From our perspective, on one hand our work presents a detailed experimental setup and an analysis at varying scales. Our experimental setup provides mean segmentation performance over whole datasets and complements this with deeper analyses by looking into different classes, location dependence, influences of segmentation boundaries, influence of segment sizes within one class and lastly architecture dependence. Statistical evaluations are complemented with a number of qualitative results.
> On the other hand, we agree w.r.t. deeper insights that our manuscript offers potential for improvement. We conducted an additional experiment investigating the influence of backbone depth on the performance of cue-expert-models which allows for conclusions on the architecture choice depending on the dataset characteristics. We find that shape experts benefit from deeper neural networks. It also turns out that extracting the texture cue can be achieved well by rather shallow CNNs.
>
>
> **[W2] Most of the findings [...] are valid but not particularly surprising and interesting"**
>
> *Response:* We understand that some of our findings may seem intuitive. However, we would like to emphasize that this is the first study in semantic segmentation experimentally confirming these intuitions on diverse depth levels down to pixel level, which we believe is of particular value (see also reviewer o1oW).
> Our study contributes to the fundamental understanding of how DNNs learn from image data. In particular, we provide new evidence that can further quantify intuitive and widely spread statements like "shape cues correspond heavily to semantic boundaries" which, to the best of our knowledge, have not been quantified before. In addition, our study reveals, that, from an information retrieval perspective, statements like CNNs being texture-loving do not hold, since neither shape nor texture alone clearly dominates the learning success (note that, this does not contradict with the finding of biases in pre-trained CNNs as studied in Geirhos et al.).
> Furthermore, we would like to put emphasis on the novelty of our findings with respect to transformer models. We observe that transformer architectures are better suited than CNNs to extract information from a cue or a limited cue combination - no matter whether these cues are based on shape or texture.
> However, qualitatively, i.e., in terms of the rankings of the different cue experts, there are no serious differences between CNNs and transformers (rank correlation of 0.973).
> This seems counterintuitive since transformers evaluated in context of classification are known to be more shape biased than CNNs (Tuli et al.). This indicates that in presence of a shape bias in semantic segmentation networks, this does not imply that transformers are less effective at learning from texture. We will adjust the narration in the manuscript to adjust the expectations of the reader.

---

> ### Author Response · Authors · 2024-11-24
> **Official Comment by Authors (2/2)**
>
> **[Q1] Using Voronoi diagrams, assigned randomly to semantic classes [...]. How about assigning semantic classes to Voronoi diagrams by their frequency in the original dataset?**
>
> *Response:* We intentionally decided to fill the Voronoi diagrams with a uniform class distribution to prevent a bias towards more frequently occurring classes during training of the cue expert and allow to learn all textures equally well. This enabled us to exclude the class frequency as a root cause for the specialization of the texture expert on a subset of classes. Following the suggestion of the reviewer, we trained additional experts on Voronoi diagrams filled according to the original Cityscapes class frequencies, leading to a worse performance when evaluating on all cues, i.e., original Cityscapes. On class level, the classes "road", "wall", "car" and "train" gained performance whereas the classes "sky", "bicycle", "traffic sign", "bus" and "sidewalk" loose the most performance measured in terms of IoU.
>
> class distribution | T_V | T_HS | T_RGB
> -------- | ------- | ----- | ----
> Cityscapes | 14.00 $\pm$ 1.74 | 19.04 $\pm$ 2.12 | 17.53 $\pm$ 1.03
> uniform  | 17.85 $\pm$ 1.30 | 20.63 $\pm$ 1.41 | 20.10 $\pm$ 0.98
>
> The ranking within the different texture experts trained on Voronoi diagrams filled according to the Cityscapes class distribution does not change, however due to the reduced overall performance T_HS and T_RBG drop one position in the ranking, respectively. The general findings for the cue influence on different semantic classes are not affected by the class distribution.
>
>
> **[Q2] The reliance on an external edge detector for shape cues introduces external biases. [...] How about using low-level filter-based edge detectors?**
>
> *Response:* We share the reviewer’s concern that edge detectors potentially introduce biases. We decided on up to three different types of shape extraction methods to disentangle cues as much as possible: a learned contour extraction method, a PDE-based low-pass filter and for the synthetic dataset a modified texture rendering. We chose the AI-based HED contour detection method over traditional filter-based edge detectors, since the latter tend to capture more texture information than HED (cf. Harary et al., Figure 3). To address the issue of semantic biases, we also applied the PDE-based low-pass filter, Edge Enhancing Diffusion (EED), as an additional method that extracts shape information of images and removes texture information (cf. S_EED-RGB expert). This method solely operates on the image information. We refer to the appendix for details on the EED method.
>
> **[Q3] When shape and color are combined (no texture), the input image becomes piecewise constant and therefore simplifies the task. The authors present this result as surprising, yet it is somewhat intuitive.**
>
> *Response:* Note that shape images with reduced color information (S_EED-HS and S_EED-V) have the same characteristic of being approximately piecewise constant. Here, we found it surprising that the models trained on these datasets with decreased color information perform significantly worse than the ones trained on S_EED-RGB with full color information. We acknowledge that our formulation has been ambiguous but have clarified that point.
>
> **[Q4] mIoU may not [...] fully capture the performance across classes with different frequencies.**
>
> *Response:* We have addressed this issue by studying frequency-weighted IoU (fwIoU) which has no significant influence on the general order of the results for the real-world dataset. We calculated the rank correlation between mIoU and fwIoU which results in a very high correlation with a Pearson correlation coefficient of 0.965 to 0.99 for the real-world dataset. For CARLA, we observe a lower rank correlation of 0.81 and see that the texture has an increased cue influence which aligns with our findings that the T cue is mostly valuable for larger segments. We included the results in the appendix.
>
> **[Q5] Figures 10 and 11 are interesting and important. I suggest moving them to the main paper.**
>
> *Response:* Thank you for recognizing the importance of Figures 10 and 11; we appreciate your suggestion to include them in the main paper. However, page constraints and an appropriate scale for readability make it challenging. We will explore ways to integrate them in the non-blind version. Either way, we enhanced their visibility by referencing them and summarizing key insights in the main body.
>
>
> **References:**
> - Sivan Harary et al. "Unsupervised domain generalization by learning a bridge across domains." CVPR, 2022.
> - Robert Geirhos et al. "ImageNet-trained CNNs are biased towards texture; increasing shape bias improves accuracy and robustness." ICLR, 2018.
> - Shikhar Tuli et al. "Are Convolutional Neural Networks or Transformers more like human vision?", arXiv:2105.07197, 2021.

---

> > ### Comment · Reviewer_bPD6 · 2024-11-26
> > **Additional discussion**
> >
> > Thank you for your response. While it partially addressed my concerns about the experimental setup, I still find the manuscript lacking in-depth and meaningful insights. I greatly value experimental and analytical studies that thoroughly investigate fundamental problems—some of my own papers also fall into this category. However, such works must present a clear perspective or thesis. This paper, however, feels more like a straightforward experimental report, and it is not clear what key message or "punchline" the authors aim to convey. After reviewing the rebuttal, my assessment remains unchanged.

---

### Author Response · Authors · 2024-11-30

We thank the reviewers for their comments on our response. We understood that the reviewers acknowledge the importance of our study however where not too surprised by our results. In the CV community however, there is the widespread word of "texture biased CNN". This of course has solid experimental foundation when massively changing the texture cue during inference. As we show consistently throughout our experiments, in training on real data the shape cue combined with color is most influential. We hope that this experimental insight is of value for the CV community to interpret CNN and ViT training.
Another criticism was the in depth investigations of the phenomena observed in our experiments. In the rebuttal we provided further studies on pixel and architecture level which was not addressed in the reviewers' responses but provide meaningful insights. As demonstrated in Figure 6, 12 and 14, cue decomposition allows for a deeper understanding of where which cue has the highest influence. We argue, that by fusing information of different experts, the cause of potential failures is more interpretable. This understanding is particularly useful when shape and texture do not correspond, e.g., a giraffe without spots like Kipekee or a fancy car with strange shape like the cybertruck. There is a high potential that these are still segmented correctly with the help of the experts which are not affected by the shift like the shape expert for a giraffe. Motivated by the reviews, we propose an uncertainty measure and its qualitative evaluation in the last paragraph of "Cue Influence Dependent on Location in an Image" in section 4.2. We kindly invite the referees to consider this and comment.

Finally, we would like to emphasize, that proving intuitions might seem to lack in-depth insights but is important since intuition can turn out to be (partially) incorrect and therefore their confirmation has a broader impact for the community.

---

### Meta-Review · Area_Chair_xPUU · 2024-12-23

**Metareview:**

This paper studies the importance of different image cues like color, shape, texture, etc., for the learning of deep semantic segmentation models. The manuscript was reviewed by four experts in the field. The recommendations are (2 x "5: marginally below the acceptance threshold", 2 x "6: marginally above the acceptance threshold"). The reviewers raised many concerns regarding the paper, e.g., unclear motivation and statement without in-depth analysis, unconvincing experimental evaluation results, etc. Considering the reviewers' concerns, we regret that the paper cannot be recommended for acceptance at this time. The authors are encouraged to consider the reviewers' comments when revising the paper for submission elsewhere.

**Additional Comments On Reviewer Discussion:**

Reviewers mainly hold concerns regarding unclear motivation and statement (e.g., lack of in-depth and detailed analysis, no new insights) (Reviewer bPD6, pTje, o1oW), unconvincing experimental evaluation results (e.g., size on the performance of texture and shape experts, results are unclear and not surprising, validity of experiments) (Reviewer o1oW, j3xz). The authors' rebuttal could not fully address the above-listed concerns.

---

### Decision · Program_Chairs · 2025-01-22

Reject